# Human olfactory-auditory integration requires phase synchrony between sensory cortices

Guangyu Zhou [1], Gregory Lane [1], Torben Noto [1], Ghazaleh Arabkheradmand[1], Jay A. Gottfried[2,3], Stephan U. Schuele[1], Joshua M. Rosenow[4], Jonas K. Olofsson[5], Donald A. Wilson[6,7] & Christina Zelano[1]

Multisensory integration is particularly important in the human olfactory system, which is highly dependent on non-olfactory cues, yet its underlying neural mechanisms are not well understood. In this study, we use intracranial electroencephalography techniques to record neural activity in auditory and olfactory cortices during an auditory-olfactory matching task. Spoken cues evoke phase locking between low frequency oscillations in auditory and olfactory cortices prior to odor arrival. This phase synchrony occurs only when the participant's later response is correct. Furthermore, the phase of low frequency oscillations in both auditory and olfactory cortical areas couples to the amplitude of high-frequency oscillations in olfactory cortex during correct trials. These findings suggest that phase synchrony is a fundamental mechanism for integrating cross-modal odor processing and highlight an important role for primary olfactory cortical areas in multisensory integration with the olfactory system.

[1] Department of Neurology, Northwestern University Feinberg School of Medicine, Chicago, IL 60611, USA. [2] Department of Psychology, University of Pennsylvania, Philadelphia, PA 19104, USA. [3] Department of Neurology, University of Pennsylvania, Philadelphia, PA 19104, USA. [4] Department of Neurosurgery, Northwestern University Feinberg School of Medicine, Chicago, IL 60611, USA. [5] Department of Psychology, Stockholm University, SE-106 91, Stockholm, Sweden. [6] Department of Child and Adolescent Psychiatry, New York University School of Medicine, New York, NY 10016, USA. [7] Department of Neuroscience and Physiology, New York University, New York, NY 10016, USA. Correspondence and requests for materials should be addressed to G.Z. (email: guangyu.zhou@northwestern.edu) or to C.Z. (email: c-zelano@northwestern.edu)

The human brain has a remarkable capacity for responding to environmental odors at minute concentrations[1]. However, adaptive olfactory-guided behaviors depend on integrating olfactory and non-olfactory cues; potent smells such as Limburger cheese may be appetizing only in the appropriate context. A long-held notion is that the human olfactory system, with its primitive cortical architecture, is especially dependent on auditory or visual information, which better allows for spatial localization and precise identification of the odor sources[2]. Indeed, recognition of odors is severely compromised without multisensory cue integration[3]. Olfaction may thus provide a suitable framework to study multisensory processing and how it confers advantages in understanding, navigating, and perceiving our environment, allowing for complex engagement with our surroundings.

Substantial progress has been made in understanding integration of sensory information in the auditory, visual, and somatosensory domains[4]. Several cortical brain areas have been identified that may integrate information from multiple primary sensory areas[5], in support of the classical hierarchical model of multisensory integration. According to this framework, unimodal inputs converge onto higher multisensory areas, which integrate multimodal information to guide decision making and behavior[6]. The superior temporal sulcus (STS)[7], lateral occipital–temporal cortex[8], posterior parietal cortex[9], and ventrolateral frontal cortex[10] have all been implicated in higher cortical multisensory processing. For example, auditory and visual information relies on STS to form an integrated representation of an action[11]. However, more recent studies have also found involvement of primary sensory areas in multisensory processing[12–14], indicating that multisensory integration may involve more distributed neural networks beyond classic hierarchical multisensory-specific areas, including primary sensory cortices[15].

A growing body of evidence suggests a key role for synchronized oscillatory activity during multisensory integration[16,17]. Coherent oscillatory firing patterns have been proposed to mediate integration and information selection across distributed neural networks[18,19]. More recent studies have begun to demonstrate similar mechanisms during multisensory integration in primates[20], including modulation of phase dynamics in primary sensory areas by cross-modal inputs.

Here we combined human intracranial electroencephalography (iEEG) methods with an auditory-word-cued olfactory matching task to test three main hypotheses about multisensory integration within the olfactory system. First, data from rodents[14], primates[21], and humans[22] suggest that cross-modal information relating to odor can modulate olfactory responses in primary olfactory (piriform) cortex (PC). Therefore, we tested the hypothesis that odor-predictive auditory cues would generate amplitude increases in PC in advance of odor stimulation, establishing that auditory stimuli alone can activate primary olfactory cortex[17]. Second, prior work suggests that synchronized oscillations reflect communication across distributed networks[23,24]. Therefore, we next tested the hypothesis that successful integration of information from auditory and olfactory domains would depend on phase synchronization of low-frequency oscillations in auditory and olfactory cortices. Third, phase–amplitude coupling (PAC) has been suggested to underlie distributed cognitive function, facilitating local computations[36,37,70]. Therefore, we tested the hypothesis that synchronized low-frequency oscillations in auditory cortex would couple with higher frequency oscillations locally in PC as a potential means of information from one modality impacting local computations in primary cortex of another modality. We found that spoken cues evoked phase locking between low-frequency oscillations in auditory and olfactory cortices prior to odor arrival. This phase synchrony occurred only when the participant's later response was correct. Furthermore, the phase of low-frequency oscillations in both auditory and olfactory cortical areas coupled to the amplitude of high-frequency oscillations in olfactory cortex prior to odor arrival during correct trials. These findings suggest that phase synchrony is a fundamental mechanism for integrating cross-modal odor processing and highlight an important role for primary olfactory cortical areas in multisensory integration with the olfactory system.

## Results

**Experimental design**. To examine olfactory and auditory responses in the human brain, we recorded iEEG local field potentials (LFPs) from seven participants who took part in a sound-cued odor identification task (Fig. 1a). Each trial began with a computerized, spoken descriptive word (rose or mint), followed several seconds later by the presentation of an odor. The auditory cues and the odor did not overlap. Following odor sampling, participants indicated whether the odor matched the cue. Spectrotemporal responses to the auditory cues prior to odor arrival in auditory and olfactory regions of interest, including auditory cortical areas surrounding the superior temporal gyrus (STG) and the STS (Fig. 1b) and PC (Fig. 1c), were examined using time–frequency analysis.

**Auditory cues increase delta and theta amplitude in PC**. To test our first hypothesis that odor-predictive auditory cues would generate power increases in PC in advance of odor stimulation, we computed spectrograms aligned to the auditory cue onset in our regions of interest. We found LFP amplitude modulations in both auditory and olfactory cortical brain regions (Fig. 2a). As expected, auditory cues induced amplitude increases in auditory cortex, presumably reflecting the initial encoding of the sounds. Specifically, LFP amplitudes increased in the delta–theta (1–7 Hz) and gamma (36–200 Hz) ranges (false discovery rate (FDR) corrected for multiple comparisons $p < 0.05$; max $z = 14.7$, permutation test) (Fig. 2a, b). Maximal amplitude increases in response to the spoken word cues consistently occurred in STG in each individual (Figs. 1b and 2a). In agreement with other studies examining LFPs in auditory cortex[25–27], we also found LFP suppression in alpha and beta bands (7–30 Hz) (Fig. 2a, b). Cue-evoked responses were consistently evident in auditory cortex at the individual level in each participant's spectrogram (Supplementary Figure 1a).

As hypothesized, auditory cues also evoked responses in PC before any odor was presented (FDR corrected $p < 0.05$; max $z = 7.46$, permutation test) (Fig. 2a). Specifically, auditory cues induced LFP amplitude increases in the delta and theta ranges. In contrast to auditory cortex, cue-evoked PC responses were restricted to the lower frequency ranges (between 1 and 7 Hz), with a peak frequency at 4.72 Hz (Fig. 2b). There were no consistent cue-evoked increases in higher frequency oscillatory amplitudes in PC (>8 Hz). This effect of low frequency amplitude increases was also present at the individual level in each participant's spectrogram ($t(6) = 11.82$, $p = 2.21e-5$, paired $t$ test; Fig. 2c and Supplementary Figure 1a, b) and at the individual single-trial level in non-baseline-corrected single-trial amplitude time series (Fig. 2d). Across participants, the frequency of the peak response ranged from 1.17 to 5.25 Hz. Auditory cue-evoked responses occurred first in auditory cortex, followed by PC. Cue-induced, low frequency responses in auditory cortex peaked at 0.34 s compared to 0.64 s in PC ($p < 0.0001$, $z = 8.81$, permutation test; Fig. 2e, f). Thus auditory stimuli that predicted ensuing odors evoked responses in auditory cortex, followed by responses in primary olfactory cortex, all prior to the arrival of the expected odor. Integration of auditory information with olfactory

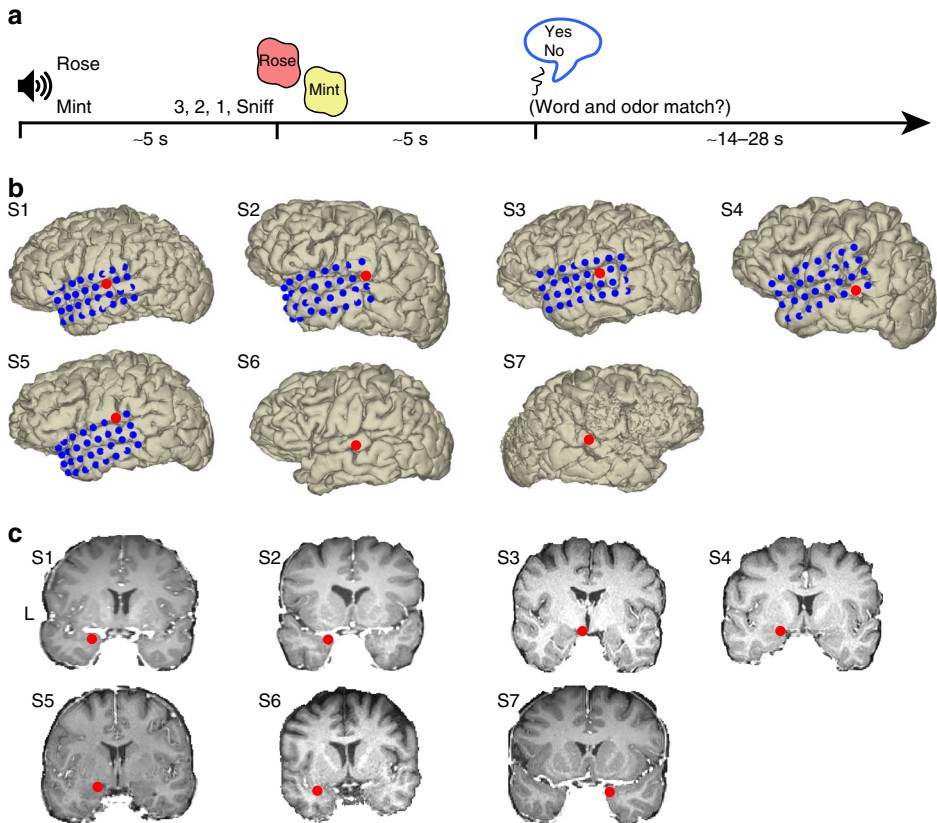

**Fig. 1** Experimental design and electrode locations. **a** Auditory–olfactory matching task. Each trial began with a computerized, spoken descriptive word (rose or mint), followed ~5 s later by the presentation of an odor. After smelling the odor, participants verbally indicated whether the odor matched the word. A trial was considered correct if the participant responded "yes" when the odor matched the cue or "no" when the odor did not match the cue. **b**, **c** Electrode locations. Red dots on individual subject brains indicate electrodes located in the superior temporal gyrus (**b**) and piriform cortex (**c**) for each participant (S1–S7). Blue dots in **b** show the entirety of implanted parietal grids (S1–S5). S6 and S7 had depth wires implanted with an electrode in superior temporal gyrus. L left hemisphere

information thus begins prior to the arrival of the olfactory stimulus. That cue-induced responses in PC were restricted to the theta range dovetails with recent findings highlighting the importance of theta band oscillations for odor coding in human PC[28], suggesting similar frequency characteristics of olfactory predictive and stimulus-evoked coding.

To confirm that these cue-evoked responses were restricted to PC and not global, we conducted two additional analyses. First, we compared cue-aligned spectrograms from PC to those in the electrodes lateral to PC on the same depth wires, averaged across participants. We did not find cue-evoked responses in these electrodes located outside of PC (Fig. 3a). Next, we conducted follow-up analyses at the individual level, by computing spectrograms from the signals recorded by every electrode on every participants' PC-bound depth wire (Fig. 3b). We found that cue-evoked low-frequency amplitude increases were largest in electrodes that were located inside PC compared to those located in non-olfactory areas (Fig. 3c). At the individual level, maximal cue-evoked low frequency (1–7 Hz) responses were significantly larger in PC compared to non-PC electrodes ($t(6) = 3.79$, $p = 0.009$, paired $t$ test; Fig. 3d). These data suggest that auditory cue-evoked responses were restricted to PC electrodes, with no significant effects in nearby non-PC electrodes.

We next wanted to confirm that cue-evoked amplitude increases in PC were driven by the cues and not simply induced by respiration. Nasal inhalation drives LFP activity in PC[29,30], and certain cognitive tasks can modulate respiratory patterns,

including imagination of odors[31]. Together, these facts suggest the possibility that, if auditory cues induced sniffs, the cue-evoked activity we observed in PC could be due to sniffing, rather than the sounds. Thus we analyzed each participant's respiratory data around the time of the cues in order to confirm that our results were not driven by nasal airflow (Fig. 3e). We first averaged the respiratory signals within each participant during 2 s time windows before and after the auditory cue. We found that neither the presence of the cue nor the identity of the cue changed respiration (two-way repeated-measures analysis of variance, effects of cue or word; all $ps > 0.39$). Across participants, there was no change in maximal airflow ($t(6) = -0.56$, $p = 0.59$, paired $t$ test) and no change in minimal airflow ($t(6) = -0.67$, $p = 0.53$, paired $t$ test) during the 2 s window before and after the cues. To be sure that the cue did not induce a change in the size of the nasal inhalation following the cue, even if it occurred outside of the 2 s time window, we compared the size of breaths preceding the cue to the size of the next breath following the cue. Breaths taken following cues did not differ from those taken before the cue in terms of peak inhale airflow ($t(6) = 1.17$, $p = 0.29$, paired $t$ test), peak exhale airflow ($t(6) = -0.59$, $p = 0.58$, paired $t$ test), or inhale volume ($t(6) = 1.53$, $p = 0.18$, paired $t$ test). Our analysis confirmed that the auditory cues did not change participants' respiratory behavior, and therefore our results were not driven by nasal airflow.

**Phase synchronization between auditory cortex and PC.** Participants were required to match the odors to the spoken words,

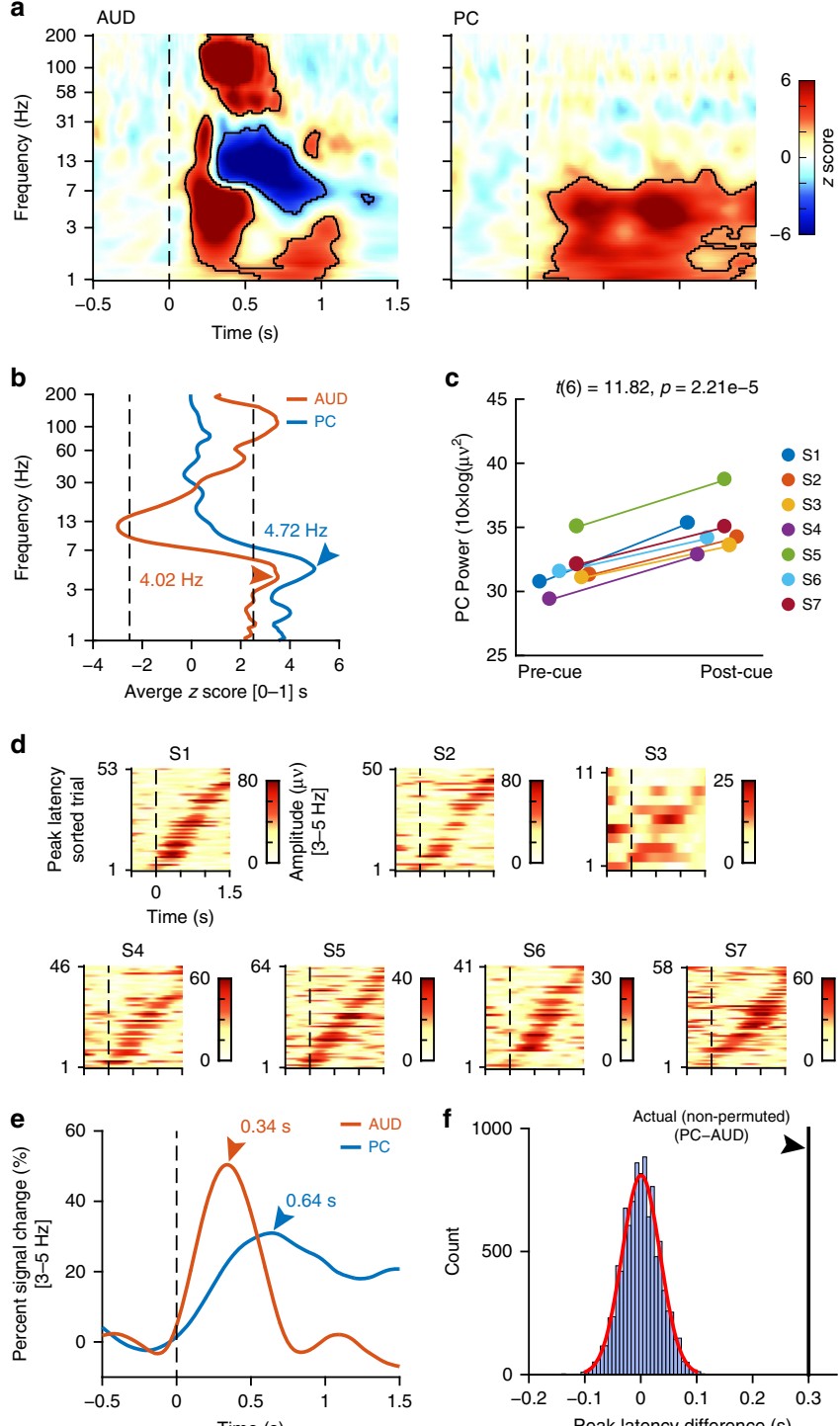

**Fig. 2** Cue-induced local field potential (LFP) amplitude changes. **a** Group-level auditory-cue-aligned spectrograms were computed from auditory cortex (AUD) and piriform cortex (PC) LFPs. Areas of statistical significance are outlined in black (false discovery rate (FDR) corrected $p < 0.05$, permutation test). **b** Frequency of maximal cue-evoked amplitude modulations in AUD (red) and PC (blue). The average $z$ scores over a time window of [0−1] s following auditory cues are plotted as a function of the frequency. Arrows indicate peak frequencies. The vertical black dotted lines indicate the FDR-corrected threshold for statistical significance in **a**. **c** Individual-level analysis of auditory cue-induced responses in PC. Pre-cue and post-cue average low frequency (1–7 Hz) amplitudes are shown for each participant (S1–S7). **d** Single-trial theta (3–5 Hz) amplitude time-series for each individual participant (S1–S7). On each plot, trials are sorted by latency-to-peak. These plots show raw data that has not been baseline corrected. **e** Percentage of signal change at peak response frequency (3–5 Hz) in AUD (red) and PC (blue). Arrows indicate the time from auditory cue onset ($t = 0$) to the peak response. **f** Peak latency difference between AUD and PC. The histogram (blue bars) indicates the null distribution of permuted differences between AUD and PC latencies. The red line indicates the normal curve fit. The vertical black line represents the actual (non-permuted) peak latency difference (PC−AUD), revealing a statistically significant difference ($p < 0.0001$, $z = 8.81$, permutation test)

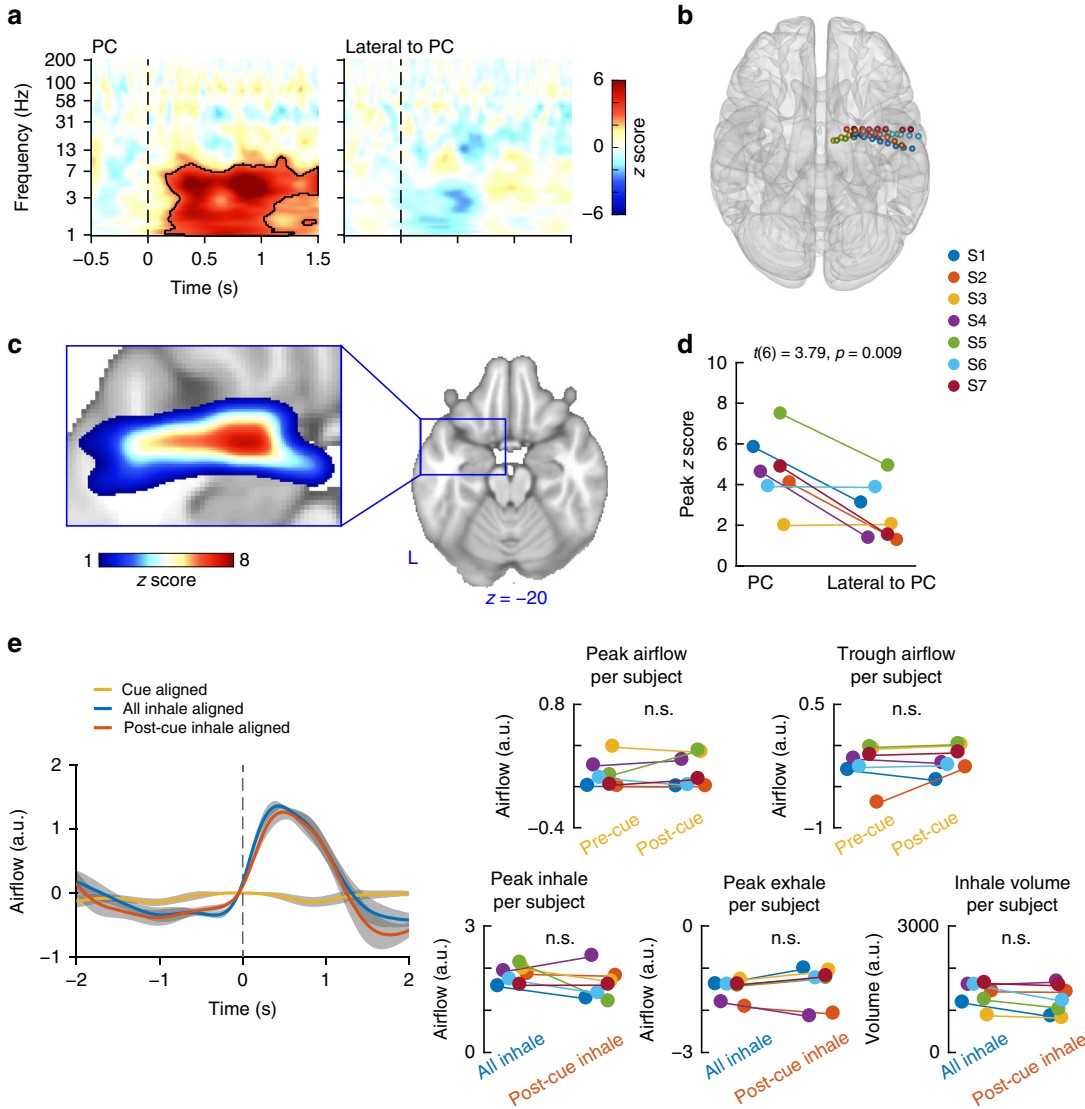

**Fig. 3** Control analyses. **a** Auditory cue-induced responses in piriform cortex (PC) were anatomically specific. Cue-aligned spectrograms for electrodes inside and lateral to PC. Black-outlined clusters indicate statistical significance (false discovery rate corrected $p < 0.05$, permutation test). Dashed lines indicate the onset of the auditory cue. **b** Location of PC-bound depth wires for all participants overlaid on the Montreal Neurological Institute standard brain. Each dot represents one electrode along the depth wire for each participant. One patient who had right hemisphere placement of the PC depth wire was mirrored to the left hemisphere. **c** Spatial distribution of auditory cue responses showing a hot spot in PC. Hotter colors represent larger response magnitudes that were determined by the peak response following auditory cues, computed separately for all electrodes along PC-bound depth wires for all participants. **d** Responses in depth wires located inside PC were consistently larger than those located outside of PC. Values are shown for each participant (S1–S7) and were compared using a two-tailed paired $t$ test across participants ($t(6) = 3.79$, $p = 0.009$). **e** Auditory cue-induced responses in PC were not driven by respiratory modulations. Auditory cues did not modify respiratory behavior. Raw respiratory signals were aligned to the auditory cue and then averaged, showing no change in breathing following auditory cues (yellow line). Respiratory data aligned to all inhale onsets over the entire experiment (blue line) and the subset of inhale onsets that occurred after the auditory cue and before the odor (red line), show that the cues also did not impact the subsequent breath. Gray shaded areas surrounding each line indicate standard error over participants. The panels on the right show individual participants' maximal and minimal airflow values during the pre- and post-cue time windows (top), individual participants' inhale and exhale peaks (bottom left and middle), and individual participants' inhale volumes for breaths taken before and after the cues (bottom right); n.s. indicates $p > 0.05$, two-tailed paired $t$ test

and integration of information from the two modalities was thus required for task performance. Low-frequency oscillations have been suggested as a mechanism of functional connectivity across distributed neural networks[32]. Based on this, we hypothesized that, following the cue and prior to the presentation of the odor, low-frequency LFP oscillations in auditory and olfactory cortices would become phase locked. To this end, we estimated the strength of phase locking across frequencies from 1 to 30 Hz between auditory cortex and PC following the cue, using the

Phase-Locking-Value index (PLV)[33] (Fig. 4). At the individual level, we found consistent, robust auditory–PC phase locking following the cue in each participant in the low-frequency ranges (FDR corrected $p < 0.05$; max Rayleigh's $z = 30$; Fig. 4a). A paired $t$ test revealed that post-cue PLV is statistically higher than pre-cue baseline PLV ($t(5) = 6.44$, $p = 0.0013$; Fig. 4b). These results confirm that olfactory-relevant auditory cues induced phase synchronization between PC and auditory cortex prior to the arrival of the odor.

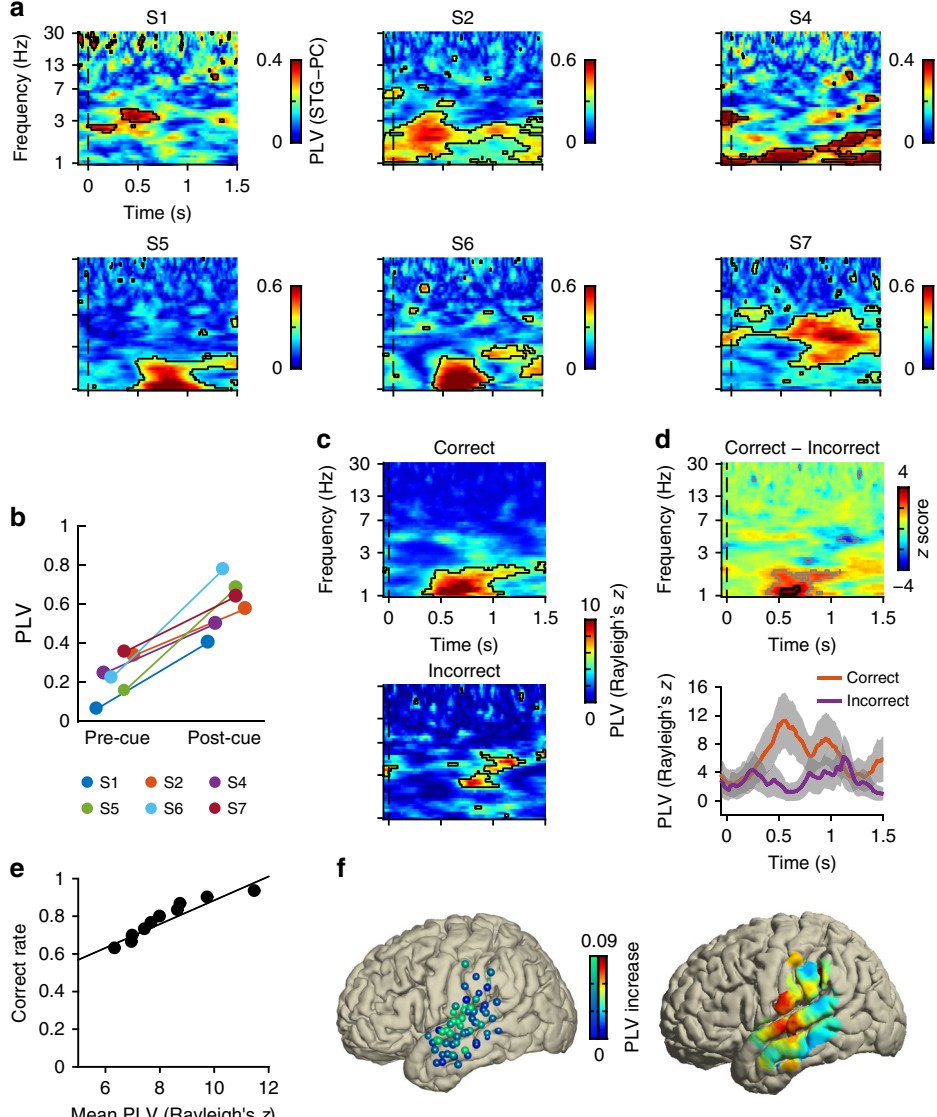

**Fig. 4** Auditory cue-induced phase synchronization between auditory and olfactory cortices. **a** Raw phase-locking value (PLV) computed between auditory cortex (STG) and piriform cortex (PC) is shown for each participant (S1–S2, S4–S7). Black outlined areas indicate statistically significant clusters (false discovery rate (FDR) corrected $p < 0.05$, Rayleigh test). The dashed line indicates the onset of the auditory cue. Note that we did not compute PLV for S3 because this participant did not complete enough trials for individual-level PLV analysis (see Methods). **b** Auditory cue-induced PLV increases at the individual participant level. PLV strength before and after auditory cues was compared using a two-tailed paired $t$ test ($t(5) = 6.44$, $p = 0.0013$). **c** Phase synchronization between primary auditory and olfactory cortices was stronger during correct compared to incorrect trials. PLV is shown computed for correct (top) and incorrect (bottom) trials separately. Dashed lines indicate the onset of auditory cues. **d** Phase synchrony was stronger when the participant's response was correct. Direct statistical comparison between correct and incorrect trials is shown in the upper panel. Black outlines indicate statistically significant clusters (FDR corrected $p < 0.05$, permutation test) and gray outlines indicate statistically significant clusters at uncorrected $p < 0.05$. The lower panel shows the averaged PLV time series at the peak frequency. Gray shaded areas denote the standard error obtained through bootstrapping. **e** PLV strength predicts future accuracy. Bootstrapped PLV values are shown plotted against the mean accuracy of the subset of trials included in each repetition. The strength of PLV for each subset of trials strongly correlates with the future behavioral accuracy ($r = 0.943$, $p = 0.000043$, Pearson's correlation). **f** Auditory cue-induced PLV is maximal between PC and STG. All parietal electrodes for all participants are shown as dots overlaid on the standard Montreal Neurological Institute (MNI) brain. On the left, each dot represents one electrode, color-coded by the strength of cue-induced PLV between PC and that particular electrode; greener colors indicate stronger PLV. On the right, raw PLV values are shown interpolated into a heat map overlaid on the standard MNI brain surface; warmer colors indicate stronger PLV

If auditory–PC phase synchronization is important for information transfer between auditory and olfactory cortices, we would expect that it should be stronger during trials when cross-modal information was successfully integrated. Therefore, we tested the hypothesis that auditory–PC phase locking was mainly present during trials in which the participant's response was correct. We pooled all trials (252 correct and 71 incorrect)

from all participants and computed PLV for correct and incorrect trials separately. To account for the difference in the number of correct and incorrect trials, we used a resampling method. In each bootstrap repetition, 71 out of the 252 correct trials were resampled without replacement, and the PLV was calculated from this subset of trials. The average PLV over 200 repetitions was taken as the PLV for correct trials. Visual inspection of these

PLV maps revealed that low frequency (1–3 Hz) auditory–PC phase synchronization was only present during correct trials (FDR corrected $p < 0.05$; max Rayleigh's $z = 12.5$; Fig. 4c). In a direct statistical comparison between the two conditions, cue-evoked phase locking was significantly stronger during correct trials, compared to incorrect trials (FDR corrected $p < 0.05$; max $z = 4.27$, permutation test; Fig. 4d).

While the finding of stronger PLV during correct compared to incorrect trials supports the hypothesis that multisensory integration requires phase synchrony, this analysis would be strengthened by moving beyond the binary classification of trials. If phase synchrony between auditory and olfactory cortices is required for integration, we would expect to find that the strength of PLV should predict the accuracy of the response. To test this hypothesis, we used a bootstrapping technique, since PLV is a cross-trial measure and cannot be computed at the single-trial level. We pooled all trials (correct and incorrect), from all subjects, into a single vector from which we randomly selected a subset of trials, and we repeated this selection 10,000 times. On each repetition, we computed the PLV for that particular subset of trials, and we computed the proportion of correct trials within that subset. This allowed us to ask whether the PLV strength for a given subset of trials could predict the behavioral accuracy for that subset of trials. We found a strong positive correlation between PLV strength and behavioral accuracy ($r = 0.943$, $p = 0.000043$, Pearson's correlation; Fig. 4e). To assess the stability of this result, we performed the bootstrapping analysis 1000 additional times, generating a distribution of correlation $r$ values. We found that the result was stable across repetitions, confirming the statistical significance of the correlation (95% confidence interval: [0.9407, 0.9502], bootstrapping method). These findings strongly suggest that oscillatory phase synchronization between cross-modal sensory cortices, prior to the arrival of an odor, is a marker of successful multisensory integration between the auditory and olfactory systems.

Thus far, we focused our phase-synchronization analysis on coupling between PC and the auditory cortical electrode that showed the strongest response to the spoken words. Auditory cortex comprises a broad area of the STG and the STS, and studies indicate functional heterogeneity across these areas[34]. To test the spatial specificity of the observed phase synchronization between auditory cortex and PC, we computed PLV between PC and all electrodes in the auditory cortex including STG and STS in all seven of our participants. We found that PLV was maximal in areas corresponding to voice-selective auditory cortex[34] (Fig. 4f), with a hot spot of maximal PC phase synchronization roughly corresponding to Brodmann areas 41 and 42.

Taken together, these data highlight the importance of primary olfactory cortex in the integration of cross-modal information within the olfactory system. They further suggest that oscillatory phase synchronization may be an important component of olfactory multisensory integration and corroborate a large body of literature indicating the importance of oscillatory neural activity in olfactory processing[28,35].

**PAC between auditory cortex and PC.** Phase synchronization of low-frequency oscillations between auditory and olfactory cortices is consistent with the established idea that low-frequency oscillations underlie functional connectivity across distributed networks[16]. However, high-frequency oscillations are thought to reflect local computations, including sensory processing. Coupling between the phase of low-frequency oscillations and the amplitude of higher-frequency oscillations is thought to be involved in cognitive processing, thus providing a means by which network coupling can impact local computations[36]. In line

with this, PAC has been linked to attention and learning[37]. Though little is known about PAC in olfactory brain areas, a recent study suggests that, during rest, there is inherent ongoing coupling between theta phase and beta amplitude in human PC[30]. This suggests that PAC in these particular frequency bands may be of particular importance for odor coding in the human brain. To determine whether odor-predictive cues modulate theta–beta coupling, we computed comodulograms of the modulation index (MI)[38] with low-frequency phase ranging from 1 to 13 Hz and high-frequency phase ranging from 13 to 200 Hz (Fig. 5a). We found two major clusters of PAC within PC, including delta–gamma and theta–beta, both of which have been previously reported in human PC[30].

To determine whether PAC in these frequency ranges was affected by the auditory cues, we computed comodulograms separately for time windows preceding and following cues (Fig. 5b). Interestingly, a comparison of MI between pre-cue ($-5$–0 s before cue) and post-cue (0–5 s after cue) revealed that the cues strongly increased theta–beta and theta–gamma coupling within PC (FDR corrected $p < 0.05$; max $|z| = 14.36$, permutation test). There was also a modest decrease in low delta–low gamma coupling, suggesting that auditory cues might also have increased the frequency of maximal PAC in PC.

We next hypothesized that this cue-induced PAC in PC might be related to coupling with low-frequency oscillations in auditory cortex, which were phase-locked to the same in PC, suggesting that, beyond our PLV findings, auditory responses could impact the timing of higher-frequency oscillations in PC as well. To test this hypothesis, we computed the MI between the phase of low-frequency oscillations in auditory cortex and the amplitude of high-frequency oscillations in PC. This analysis revealed coupling between the phase of auditory theta oscillations and the amplitude of PC beta oscillations and between auditory delta phase and PC gamma amplitude (Fig. 5c). However, when examining these effects following the presentation of cues, we found that only the auditory theta and PC beta coupling was increased during the post-cue time window (FDR corrected $p < 0.05$; max $|z| = 7.25$, permutation test, Fig. 5d). These data suggest that cross-modal cues induce cross-regional PAC that increases the strength of existing intrinsic coupling dynamics in olfactory cortex.

We found theta–beta coupling both within PC and between auditory cortex and PC. We next asked whether both of these forms of PAC were related to integration during the task or whether one of them dominated in terms of task relevance. We thus computed MI using correct and incorrect trials separately, for both types of coupling (within PC and between cortices). We found that only cross-regional PAC was stronger during correct versus incorrect trials (FDR corrected $p < 0.05$, max $|z| = 5.03$, permutation test, Fig. 5e), with no significant differences between correct and incorrect trials for within-PC coupling (FDR corrected $p > 0.05$, max $|z| = 3.17$, permutation test, Fig. 5f). These results suggest that the modulation of PC beta oscillations by STG theta oscillations is important for integration of auditory and olfactory information and highlight an important role for cross-regional PAC in multisensory integration.

Taken together, these data suggest that olfactory-relevant spoken word cues induce a phase shift in low-frequency PC oscillations and also modulate the amplitude of PC beta oscillations, thus impacting the timing of local computations in olfactory cortex. Notably, we found no differences in overall LFP amplitude in any frequency between correct and incorrect trials (FDR corrected $p > 0.05$, permutation test), suggesting that phase dynamics, rather than LFP amplitude modulations, are related to the cognitive demands of integrating auditory and olfactory information.

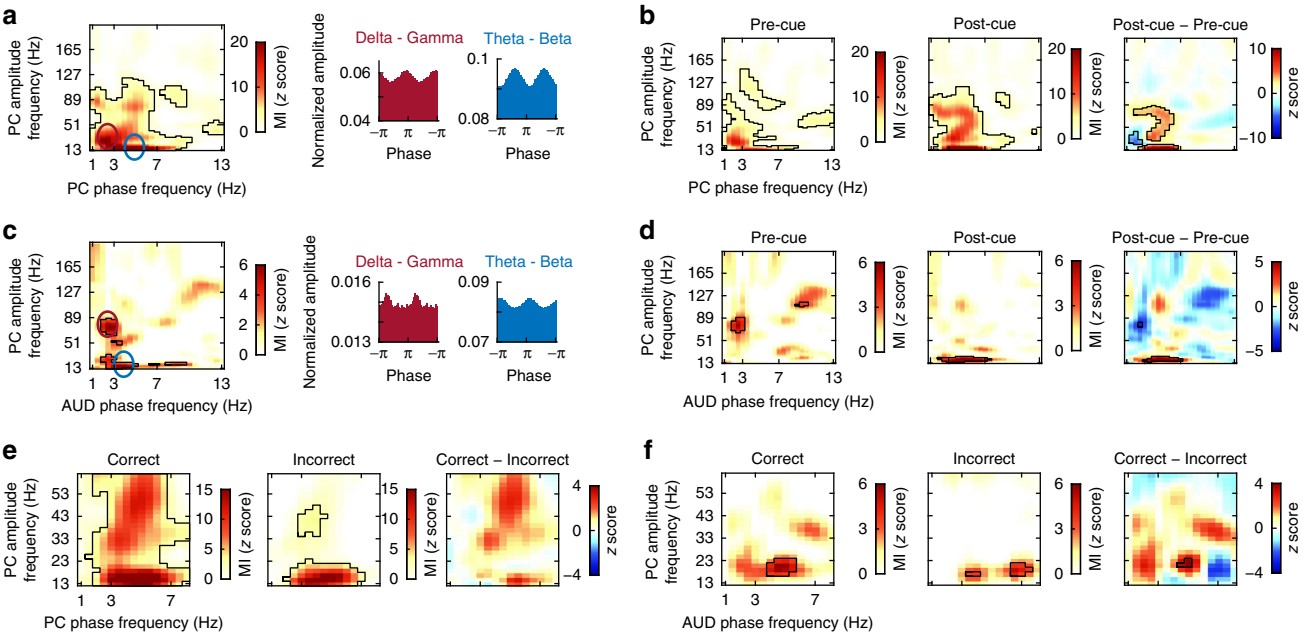

**Fig. 5** Auditory cues induced cross-frequency coupling. **a** On the left, group-level comodulograms of within-piriform cortex (PC) cross-frequency coupling are shown. Values on the plot represent the modulation index (MI) for each phase amplitude pair (see Methods). On the right, phase–amplitude distributions of the maximal modulatory frequency are shown (corresponding to red and blue circles on the left panel). **b** Auditory cue-induced changes in phase–amplitude coupling in PC. In the left and middle columns, comodulograms computed from pre-cue ([−5, 0] s) and post-cue ([0, 5] s) time windows are shown separately, with the difference between the two on the far right. **c** On the left, group-level comodulograms of between auditory (AUD)–piriform cross-frequency coupling are shown. Values on the plot represent the MI for each phase amplitude pair (see Methods). On the right, phase–amplitude distributions of the maximal modulatory frequency are shown (corresponding to red and blue circles on the left panel). **d** Auditory cue-induced changes in phase–amplitude coupling between AUD and PC. In the left and middle columns, comodulograms computed from pre-cue ([−5, 0] s) and post-cue ([0, 5] s) time windows are shown separately, with the difference between the two on the far right. **e, f** Phase-amplitude coupling between AUD and PC amplitude is stronger during correct trials. Comodulograms computed separately for correct and incorrect trials for both within-PC coupling (left) and AUD-PC coupling (right) are shown. Increased modulation during correct compared to incorrect trials was only evident for AUD–PC coupling in the theta–beta ranges. In all instances, comodulogram z scores were corrected for multiple comparisons using false discovery rate ($p < 0.05$, permutation test)

## Discussion

Information about the surrounding environment is typically obtained from more than one sensory system, requiring our brains to integrate multimodal stimuli into a coherent sensory experience. This is especially important in human olfaction, a system strongly dependent on supporting multisensory input. Here we addressed an unresolved question: How does the human brain enable odor identification by integrating multisensory cues? We found that multisensory integration with the olfactory system relies on low-frequency phase shifts in primary olfactory cortex. Following spoken word cues, low-frequency oscillations in primary olfactory cortex became synchronized with those in voice-activated auditory cortex, prior to odor arrival. Although the auditory words presented in our study did not contain predictive information about the identity of the future odor, they did carry vital information necessary for proper performance of the olfactory perceptual decision. Thus information from the auditory system was required in order to make an olfactory perceptual decision. In this sense, integration was necessary for completion of the task. Our data suggest that phase dynamics across distributed networks, including primary sensory areas, reflect underlying olfactory sensory integration in the human brain. Importantly, olfactory–auditory phase synchrony was associated with accurate performance on the cross-modal task.

An important aspect of our study was that the auditory and olfactory stimuli did not arrive simultaneously; rather, auditory cues preceded the arrival of the odors. Thus we were able to examine connectivity between auditory and olfactory cortices—and activity in olfactory cortex—prior to the arrival of the odor,

allowing us to isolate the effects of integration from responses to odor stimuli. Given this experimental design, our findings relate not only to multisensory integration but also to predictive coding theory. Our data examine predictive coding from a cross-modal view point, which is consistent with typical real-world experience, where a stimulus from one modality frequently predicts the arrival of a stimulus from a different modality. For example, the sight of lightning predicts the coming sound of thunder or the sound of a clicking stove predicts the arrival of the smell of natural gas. These common, real-world situations require both multisensory integration and predictive coding. Thus our data suggest that top–down influences on human olfactory cortex may be mediated by phase synchronization as well.

Most research on multisensory interactions in the olfactory system has been conducted in rodents, with less understanding of these mechanisms in the human brain. In rodents, it has been shown that both gustatory[39] and auditory[14] stimuli modulate single-unit activity in primary olfactory areas. Tastants applied to the tongue induce responses in rodent PC[40], and auditory tones induce responses in the rodent olfactory tubercle[14]. In humans, functional magnetic resonance imaging (fMRI) studies have shown that visual and auditory stimuli that are linked to odors can activate PC[22,41,42], but the slow nature of the hemodynamic blood-oxygen-level-dependent signal prohibits full understanding of the timing and frequency composition of these responses. Specifically, fMRI does not allow characterization of oscillatory phase dynamics across different sensory cortices. By using iEEG methods, we were able not only to show that PC plays a role in multisensory integration but also to characterize the neural

oscillatory dynamics involved: Phase synchrony between auditory and olfactory cortical oscillations enhances integration of information across the two sensory systems.

Our results directly support the emerging view that multisensory integration is reliant not only on classic hierarchical multisensory areas[20] but also on phase synchronization and resetting across distributed oscillatory networks[43], further emphasizing an important role for primary sensory areas in this process. In rodents and monkeys, several studies have found primary sensory cortical involvement in multisensory integration of visual, auditory, gustatory, and somatosensory information[20,44–46]. Such prior results and those of the present study support the postulation that multisensory integration involves a distributed network with multiple redundant pathways[5,47] and that oscillatory phase synchrony may play a key role in these mechanisms[48–51]. Our finding that oscillations in primary olfactory cortex become synchronized with oscillations in auditory cortex is consistent with both of these postulations.

Our finding of synchronized low-frequency oscillations in olfactory and auditory cortex is in line with a large body of research indicating low-frequency phase synchrony as a means of coupling across neural networks[32,52,53]. These synchronized low-frequency oscillations, in turn, modulated higher-frequency oscillations in PC, including those in the beta and gamma range. Low-frequency phase modulation of gamma amplitudes in human PC is of particular interest, given a previous study suggesting a lack of systematic odor-induced gamma amplitude increases during an odor detection task[28]. In agreement with this study, we did not find consistent gamma amplitude increases in response to auditory cues. However, we did find that gamma amplitudes were modulated by low-frequency phase dynamics in primary auditory and olfactory cortices.

One limitation of our study is the use of data from patients with temporal lobe epilepsy. While human electrophysiology data can only be obtained from olfactory areas in patients requiring this type of surgery, an important consideration is whether these patients are good representatives of normal auditory–olfactory processing. These patients sometimes show a mild deficit on olfactory testing, as well as potential atrophy of the PC in patients with medial temporal sclerosis. Surgical epilepsy patients are the only setting where this study could be ethically conducted, but we openly acknowledge that the disease pathology is a potential limitation. We took several steps to minimize the impact of these limitations on our data. No patient enrolled in our study had clinically noted abnormal volume or functional activation of PC. All patients were able to reliably identify and distinguish between the odors used in the task. Importantly, no participant had a seizure focus zone comprising auditory or olfactory cortex. Finally, any trials during which epileptiform activity was present were removed from further analysis. Another limitation of intracranial techniques is limited spatial coverage. As a result of this, we included one participant whose auditory electrode was located in the right (language non-dominant), as opposed to the left hemisphere. We included this participant in our analyses despite the dominance of the left hemisphere in language representation, because though reduced, the right hemisphere is also involved in language processing[54]. Importantly, results obtained from the electrode implanted on the right side are consistent with those obtained from the electrodes implanted on the left side.

In sum, the present results suggest oscillatory dynamics reflecting a mechanism that enables the human brain to accurately match an odor to a preceding auditory cue. While there is increasing evidence of multisensory integration in early stages of sensory neocortical pathways, our study provides evidence of early integration between a neocortical (auditory) and archicortical (olfactory) system in humans, suggesting early evolution of such multisensory convergence. Apart from our objectives to reveal the multisensory-integrative processing capacity of the evolutionarily preserved olfactory system, understanding such processes may also be of clinical research interest, given that odor identification deficits constitute a very early marker of generalized cognitive impairment and dementia among older individuals[55–58]. As we learn more about the specific frequencies that enable olfactory system computations, future work may benefit from the present findings and facilitate a better understanding of how these cross-modal circuit functions are perturbed by neurodegeneration and neural pathology.

## Methods

**Participants.** Seven surgical participants (four women) with medically resistant epilepsy participated in this study. Inclusion criteria for this study included the clinical need for surgical implantation of electrodes into both olfactory and auditory cortices and the ability to identify and distinguish the odors used in this study. Any patient whose seizure focal zone included PC or a lesion in PC was excluded from the study.

The Institutional Review Board of Northwestern University approved the study, and all patients gave written informed consent to participate.

Table 1 summarizes the patients' demographic information. Their ages at surgery ranged from 25 to 36 years (average: 33 years). The average duration of epilepsy is 9.86 years (ranging from 2 to 36 years).

**Behavioral task.** Each trial began with a computer-generated spoken cue consisting of either the word rose or mint. Sounds were presented using PsychToolbox[59–61] and Matlab (MathWorks Inc., Natick, MA, USA), via a laptop placed in front of the participant. On an average of 4.8 s (ranging from 3.3 to 7.2 s) following the cue, the subject smelled the pure odor of rose (essential oil) or mint (methyl salicylate) delivered through opaque plastic squeeze bottles placed under the subject's nose by the experimenter. The participant then indicated whether or not the odor matched the cue by answering yes or no. A synchronization signal was sent to the iEEG recording system (Nihon Kohden) via Matlab using digital output from a data acquisition device (USB-1208F, Measurement Computing) in order to mark the cue onset, the respiratory cycle corresponding to the sniff onset, and the participants' response. Participants completed between 48 and 64 trials, except for 1 participant who completed only 16 trials due to clinical constraints. The average inter-trial interval was 21.3 s, ranging from 14 to 28 s, across participants. The average performance on the task was 73.3% correct (S1: 79.69%, S2: 75.56%, S3: 31.25%, S4: 91.07%, S5: 100%, S6: 87.5%, S7: 46.88%), which means the response was yes while the odor matched with the sound cue or no if they were not matched.

**Table 1 Patient demographics**

| Patient | Gender | Age (years) | Handedness | Epilepsy duration (years) | Epileptogenic zone | Brain MRI |
|---|---|---|---|---|---|---|
| S1 | M | 32 | Right | 10 | Left basal temporal | Normal |
| S2 | F | 27 | Right | 5 | Left mesial temporal | Left MTS |
| S3 | M | 47 | Left | 2 | Left temporal lobe | Normal |
| S4 | F | 29 | Right | 7 | Left temporal lobe | Normal |
| S5 | F | 25 | Right | 3 | Left mesial temporal | Normal |
| S6 | F | 36 | Right | 36 | Left mesial temporal | Left MTS |
| S7 | M | 34 | Right | 6 | Right mesial temporal | Normal |

MRI magnetic resonance imaging, MTS mesial temporal sclerosis

We found no difference in performance between the first (mean ± standard error: 73.98% ± 9.12%) and second (72.58% ± 10.72%) half of trials ($z = 0.40$, $p = 0.69$, two-tailed Wilcoxon signed-rank test). The performance was higher in cue-odor-matched trials (85% ± 7.08%) than nonmatched trials (58.17% ± 15.05%) ($z = 2.20$, $p = 0.028$, two-tailed Wilcoxon signed-rank test). However, our analyses were focused on the time period prior to odor delivery, at which time whether the odor matched the cue was still unknown.

**Respiratory and iEEG recordings**. The respiratory signal was recorded using a piezoelectric pressure transducer (Salter Labs Model #5500) attached to a nasal cannula at the participant's nose and a breathing belt placed around the abdomen. The nasal cannula signal was used for respiratory analysis, as it has a faster deflection during sniffing[62]. The respiratory signal was $z$ score normalized and respiratory features including inhale onset, peak inhale/exhale airflows, and inhale volume were calculated using *BreathMetrics*[63] for each patient.

iEEG data were recorded using the clinical Nihon Kohden system currently in place at Northwestern Memorial Hospital. The sampling rate for each participant was determined clinically and ranged from 500 to 2000 Hz across participants, with an online high-pass filter of 0.08 Hz. The reference and ground consisted of a surgically implanted electrode strip facing toward the scalp.

Electrode locations were determined using pre-operative structural MRI scans and post-operative computed tomography (CT) scans using the FMRIB Software Library's (FSL) registration tool *flirt*[64,65]. Individual CT images were registered to MRI images using a degree of freedom of 6 with a cost function of mutual information, which was followed by an affine registration with a degree of freedom of 12. Individual MRI images were registered to a standard Montreal Neurological Institute (MNI) brain (MNI152_1mm_brain included in FSL) with a degree of freedom of 12. Finally, the transformation matrices generated above were combined to create a transformation from the individual CT image to standard MNI space.

The electrodes were localized by thresholding the raw CT image and calculating the un-weighted mass center of each electrode. To account for brain shifts, grid electrodes in the auditory cortex were projected to individual brain surfaces using the method proposed by Yang and colleagues[66]. Finally, the coordinates were converted to standard MNI space using the transformation matrix generated above.

Though we analyzed spectrograms from all electrodes on the PC depth wires and all electrodes on the parietal grids, electrodes corresponding to those shown in Fig. 1 were selected by the following procedure. PC: For each subject, we first determined which subset of electrodes was anatomically located inside PC. This typically included between 1 and 3 electrodes. For the subjects who had only a single electrode in PC, we used that one. For subjects with multiple electrodes within PC, we chose the one that was closest to the center of PC. Notably, we also analyzed all piriform electrodes separately with similar results. STG: For each subject, we computed the amplitude of gamma oscillations following presentation of the auditory cue, and we used the electrode that showed the largest increase in gamma amplitude. Notably, this resulted in an electrode located in STG for each participant. We also analyzed all electrodes on the parietal grid for each participant, which can be seen in Fig. 4.

**Time–frequency analysis of iEEG data**. iEEG data were first low-pass filtered at 235 Hz followed by removal of 60 Hz line noise and its harmonics using band-stop filters at a bandwidth of 4 Hz. All signal filtering in this study was performed using two-pass zero phase finite impulse response filter as implemented in *fieldtrip* unless stated otherwise. The data were down-sampled at a sampling rate of 500 Hz and re-referenced to a common average. Then the time series was band-pass filtered at 100 log-spaced frequencies (1–200 Hz), with the bandwidth logarithmically increased from 2 to 50 Hz. The amplitude of the band-pass filtered signal was extracted using Hilbert transform and smoothed with a moving average filter kernel of 10 ms.

To evaluate event-related amplitude changes, the amplitude time series was segmented into epochs from −0.55 to 1.5 s relative to the auditory cue. For each participant, all trials were visually checked and those trials with large artifacts were removed from further analysis. Table 2 summarizes the number of good trials for each participant. The spectrogram was calculated by averaging the amplitude

epochs across trials at each frequency, which was further normalized by subtracting a baseline ([−0.55, −0.05] s relative to cue onset) average. Note that the percentage of signal change of a specific frequency band can be calculated as $100 \times (x - \text{mean} (x_{\text{baseline}}))/\text{mean}(x_{\text{baseline}})$, where $x$ is the averaged amplitude time series.

The statistical significance of the amplitude change was tested using a permutation method[67]. In each permutation, real events were shifted in time by a random amount while maintaining the relative distance between events. The modulus of the length of the time series was calculated to ensure that all shifted events were valid. Then one mean amplitude value was calculated by averaging across these events. After repeating this procedure 10,000 times, we obtained a null distribution of baseline amplitudes. To calculate the $z$-map of the real spectrum, we divide it by the standard deviation (estimated using Matlab *normfit. m*) of this null distribution. Two-tailed $p$ values were also computed from the $z$-map for multiple comparison correction using the FDR method[68].

To construct a group-level $z$-map for each region, we visually checked the $z$-map of all electrodes for each participant and chose the most responsive electrode for each region of each participant (Fig. 1; Table 2). The amplitude time series were obtained for each frequency and each participant and concatenated across participants for each region of interest. To account for inter-individual difference in the amplitude of raw signal, raw time series were $z$ score normalized before concatenating. The $z$ score spectrogram of the concatenated time series was calculated as described above.

Of note, inter-individual response peak frequencies can be identified from individual $z$-maps. To quantify cue-induced amplitude changes at the peak frequency, the baseline average amplitude and the maximal amplitude after cue were retrieved and converted into power (decibel transformed). Then the difference between baseline and post-cue maximal was tested using a two-tailed paired $t$ test.

The peak frequency of the group-level STG and PC responses were identified by averaging the $z$ score over a time window of [0, 1] s relative to auditory cue onset. Then the percentage of signal change of a narrow frequency band that enclosed both the peak frequencies of STG and PC was calculated. The maximal time point of the percentage of signal change was calculated as the peak latency for STG and PC separately. To test the significance of the peak latency difference between STG and PC, we used a permutation method. For each permutation, the trial labels for STG and PC were randomly shuffled and the permuted peak latency difference was calculated and retained. A distribution of permuted peak latency difference was obtained by repeating the above procedure 10,000 times resulting in a null distribution of peak latency difference. The mean and standard deviation of this distribution was obtained (Matlab's *normfit.m*), from which a $z$ score of the real peak latency difference was calculated.

**Phase locking value analysis**. To examine auditory cue-induced changes of the coupling between the auditory and PCs, we calculated cross-trial PLV between STG and PC. PLV is a measure of the consistency of the phase difference at a specific frequency and time between two regions over trials (Supplementary Figure 2). To obtain the phase time series, we band-pass filtered the raw time series between 1 and 30 Hz in 50 log-spaced frequencies (bandwidth: 2 Hz) for each region. Then the phase time series were obtained using the Hilbert transform method. Phase difference time series was segmented into [−0.55, 1.5] s epochs relative to cue onset. PLV together with the Rayleigh's $z$ score and $p$ value were calculated at each time–frequency point using CircStats toolbox[69].

To directly evaluate cue-induced PLV changes at the individual level, we extracted the maximal low-frequency (<7 Hz) PLV and baseline averages at the corresponding frequency. Then pre-cue baseline PLV and post-cue maximal PLV was compared using a two-tailed paired $t$ test.

Next, we calculated the PLV for correct and incorrect trials. To account for the difference in the number of correct (252) and incorrect (71) trials, the PLV for correct trials was calculated using 71 trials that were randomly drawn (without replacement) from all correct trials. This resampling procedure was repeated 200 times and the average of the resulting PLV vales was used as the final result. To compare the PLV between correct and incorrect conditions, we used a permutation method. In each permutation, 71 trials were randomly selected (without replacement) for correct and incorrect conditions, respectively, from a pool of all trials. Then a null distribution of permuted PLV differences between correct and incorrect conditions was obtained by repeating the permutation 1000 times. The mean and standard derivation of this null distribution was calculated with normal curve fitting (Matlab's *normfit.m*). Finally, a $z$ score map of the real PLV difference was calculated by subtracting the mean and then dividing by the standard derivation.

In addition to comparing the PLV between correct and incorrect directly, we examined whether behavioral performance (correct rate) was correlated with the strength of PLV. To do so, a subset of 71 trials were randomly extracted from the pool of all trials. Because the number of trials affects the PLV, we chose the number of 71, such that the resulting PLV value is comparable to those in the correct and incorrect comparison. For this subset of trials, we calculated the correct rate and average PLV within the significant time–frequency window that was obtained from correct versus incorrect comparison. This resampling procedure was repeated 10,000 times. Next, we sorted correct rates then binned them into 10 equally spaced bins, and the mean PLV was calculated for each bin. The correlation between correct rate (center of each bin) and PLV was examined using Spearman

**Table 2 MNI coordinates of regions of interest**

| Patient | Superior temporal gyrus | Piriform cortex | No. of trials |
|---|---|---|---|
| S1 | −68.6, −14.7, 7.9 | −21, 1.9, −22.5 | 53 |
| S2 | −69.2, −19.3, 13 | −15.6, 7.1, −29.4 | 50 |
| S3 | −69.2, −14.2, 8.1 | −7.5, −4.3, −13.9 | 11 |
| S4 | −69.8, −32.4, 11.9 | −19.7, 0.4, −18 | 46 |
| S5 | −65.3, −17, 17.4 | −14.8, −1.8, −18.2 | 64 |
| S6 | −67.6, −18.3, −0.1 | −26.8, 3.5, −24.6 | 41 |
| S7 | 69.5, −21.1, 15.7 | 20.2, 7, −27.9 | 58 |

*MNI* Montreal Neurological Institute

correlation analysis. To estimate a confidence interval of this correlation, the correlation analysis was performed 1000 times.

**Cross-frequency coupling analysis.** To examine the cross-frequency coupling, we adopted the phase–amplitude MI method[38]. MI is a measure of the entropy of the phase–amplitude plot obtained by binning high-frequency amplitude values by low-frequency phases. The phase time series were obtained for frequencies from 1 to 13 Hz (step: 0.5 Hz; bandwidth: 2 Hz) for STG/PC. The STG/PC amplitude time series were obtained for frequencies from 13 to 200 Hz (step: 2 Hz; bandwidth increases linearly from 4 to 50 Hz). To evaluate the PAC, the phase and amplitude time series were segmented into time windows of $[-5, 5]$ s relative to auditory cue onset. The number of bins used for MI calculation was 20. The normalized MI ($z$ score) was calculated using a surrogate method[70]. Surrogate data were generated by shifting either the phase time series or the amplitude time series using Matlab's *circshift.m* function. The MI was then calculated for the surrogate data. We calculated 200 surrogate MIs, resulting in a null distribution of MI values. The mean and standard deviation of this distribution was calculated using Matlab's *normfilt.m*. Finally, the $z$ score of the real MI was calculated by subtracting the mean of the distribution and then dividing by the standard deviation.

To examine the MI change induced by the auditory cue, the MI was calculated for pre-cue ($[-5, 0]$ s before cue) and post-cue ($[0, 5]$ s after cue), separately. The MI difference between pre- and post-cue was examined using a permutation-based method. For each permutation, the pre- and post-cue data for each trial was switched randomly. Then the MI for permuted pre- and post-cue conditions were calculated as described above. A distribution of permuted difference of normalized MI between two conditions was obtained by repeating the procedure 200 times. Thus a $z$ score and its corresponding $p$ value of the real normalized MI difference were calculated by subtracting the distribution mean from the real MI difference, then dividing by the standard deviation of the distribution. The distribution was pooled across all phase and amplitude frequencies.

The MI difference between correct and incorrected trials for the post-cue time window was examined using a similar permutation method. In each permutation, instead of switching randomly the pre- and post-cue labels for each trial, all correct and incorrect trials were pooled together, and the trial labels were shuffled. Since the number of trials differs between correct and incorrect, we used normalized MI values instead of raw MI values. A $z$ score map of the real difference in MI between two conditions were calculated from the permuted MI difference distribution as described above. Of note, multiple comparisons in the MI analysis were performed for MI and comparison between conditions, i.e., pre-cue versus post-cue and correct versus incorrect, respectively, using the FDR method.

To compare cue-induced amplitude change between correct and incorrect trials, the amplitude epochs of the concatenated time series were averaged across trials within each condition and baseline corrected. A null distribution of permuted between-condition differences was constructed by shuffling condition labels across trials. For each permutation, the amplitude change was calculated for both conditions and these were subtracted from each other. By repeating the above procedure 1000 times, we obtained a null distribution of amplitude change differences. To calculate the $z$ score of the real amplitude change, the mean of the null distribution was subtracted from it, and the result was further divided by the standard derivation of the distribution.

**Reporting summary.** Further information on experimental design is available in the Nature Research Reporting Summary linked to this article.

## Data availability

The data and code that support the findings of this study are available from the corresponding author upon reasonable request.

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

## Acknowledgements

We thank Jeremy Eagles, Enelsa Lopez, and Navid Shadlou for their technical support and assistance with data collection. This work was supported by National Institutes of Health grants from the National Institute on Deafness and Other Communication Disorders (NIDCD) (R00-DC-012803 and R01-DC-016364 to C.Z.), The National Institute of Neurological Disorders and Stroke (NINDS) (T32-NS047987 to T.N.), and The Knut and Alice Wallenberg Foundation (KAW 2016:0229 to JKO).

## Author contributions

C.Z., G.L. and G.Z., designed the study; C.Z., G.L., G.Z. and G.A. collected the data; G.Z. and C.Z. analyzed the data; T.N., S.U.S., J.A.G., J.M.R., J.K.O. and D.A.W. assisted with data collection, access to patients, and/or manuscript preparation and editing; J.M.R. performed surgeries; C.Z., G.L. and G.Z. wrote the paper.

## Additional information

**Competing interests:** The authors declare no competing interests.

