## [Peer Review File · Nature Communications]

Reviewers' Comments:

Reviewer #1:

Remarks to the Author:

General Comments

The submission is well written and organized. The results are important and the methods and results are succinctly presented, but their description leaves the reader with much ambiguity. The results are clearly positive with respect to the hypotheses presented in the Intro. The auditory cue induces oscillations in the olfactory cortex prior to odor arrival. Evidence is also presented for long-range phase synchronization between auditory and olfactory cortices at this time; and there is evidence for phase-amplitude coupling between the low frequency (theta and delta) phase and the high frequency (beta and gamma) olfactory amplitude. This is true for low-frequency oscillations in both auditory and olfactory cortices.

Despite the positive results, particularly troubling is the lack of grid recordings for subjects 6 and 7 (Fig 1). Since their auditory recordings were different from the other subjects, how were their results compared to that of the others? Also, the question of hemispheric specialization is raised – each subject was implanted in the left hemisphere, where language is represented in most subjects, except for S7. The inclusion of S7, with right hemisphere implantation Table 1, may be confounding the results.

One problem is that the manuscript is too long. I suggest that, in order to shorten it, the confirmatory results for depth strength (PC vs PC-adjacent) and respiration (and related figures) be moved to a supplementary material section.

Specific Comments

79: Change "An increasing" to "A growing".

80-82: It should be pointed out that this has been only a hypothesis before now - such patterns have been proposed to mediate ...

90: "rodents and humans" - similar results have been seen in monkeys doing cognitive saccade tasks – add reference to Babapoor-Farrokhran et al. 2017 Nature Communications. Also add reference to Meehan et al. 2017 Brain Struct Funct where an auditory cue that predicted a visual stimulus led to effects in human primary visual cortex.

92-93: This hypothesis also assumes that the cue, preceding the sensory stimuli, affects processing of the ensuing stimulus in the sensory cortex. Cite the paper by Brovelli et al. 2004 PNAS as establishing that pre-stimulus oscillatory synchrony organizes information in primary sensory cortices.

95: Phase synchrony is consistent with "communication" but does not establish that communication is enabled.

100: "driving" is NOT measured here! See next comment.

102: Phase-amplitude coupling does NOT entail any sense of "driving". To have phase at one frequency coupled to amplitude at another frequency merely says that there is coupling, not that 1 drives the other.

135: Why is there no grid for S6 & S7? Does "depth wire implanted" mean that depth wires were used in both auditory and olfactory systems for subjects S6 and S7? If so, where were the auditory electrodes implanted? Can their depth LFP recordings be compared with the ECoG recordings of the other subjects?

146: How do the spectrograms compare when the baseline is not removed? Recent LFP studies suggest that non-baseline-corrected spectrograms should also be examined.

147: How do the spectrograms compare when the baseline is not removed? Recent LFP studies

suggest that non-baseline-corrected spectrograms should also be examined.

152: Which PC electrode was used, and how was it selected?

156: The nature of this segmentation and its relevance are unclear. How long were the epochs? Is only 1 epoch displayed?

159: Is this the same as AUD shown in figure 2E? Was there even an electrode in STG for subjects 6 & 7?

161-162: The description of panel F should be clarified. What is meant by the "real peak difference"? The peak difference should vary from trial to trial. Why is it represented by a single vertical line?

171: Change "reflecting" to "presumably reflecting".

178: It is not clear that task-related LFP alpha & beta suppression corroborates prior work on auditory resting oscillations.

194: Change to "presumably reflecting".

232-233: How many depth wires were there for each patient? Did they all extend into the PC? This should be better explained - if each electrode went into the depth of the brain, how could electrode contacts be displayed on the cortical surface?

295: There cannot be 2 panel Bs in Fig 5.

314-316: This sentence should be disentangled. The first part presents a result and the second a speculation. Also, functional connectivity is not equivalent to communication. I think the former is more appropriate here.

333: Were "not matched" trials analyzed? Why or why not?

341: Should the text say "correctly matched"?

361-362: It isn't clear how voice-selective auditory cortex was identified. Specify how the contacts were selected.

364-369: The conclusions here, which are about the olfactory system DO NOT FOLLOW from the preceding text, which is about the auditory system.

372: All of the results presented in this paper have been undirected. Consequently, there is no concept of driving. Phase-amplitude coupling, also, is not directional and thus the word "drive" is inappropriate.

382: In A & C, right panels, the x-axis should be labeled. Also, the value of including the right panels is not clear. How were MI values derived?

403: It is still debatable how strongly this theory is "established".

410: What is this? Phase-amplitude coupling?

438: Change "were" to "was".

480: Remove "behavioral".

492: Change "For example" to "Specifically".

510: Is there any indication from the present results what the separate roles might be of hierarchical convergence and phase synchrony of sensory areas?

515: Even though the finding may be in line with this idea, it is important to note that the present study did NOT study communication across networks.

647: The third column is unlabelled. Also, is it the number of trials used in the analysis? If so, S3 only had 11 trials used. This is too small a number for individual-subject analyses.

651: It's not clear how the surrogate events were generated. Was the analysis at the group level?

651: It is not clear what "circularly shifting" means.

693: How were phase time series obtained? Wavelets? EMD? Hilbert Transform?

700-716: Poor English grammar in several places.

718-719: Give a better description of this method. What is MI?

727-731: Give a more complete description of the procedure used.

768: This text is too terse! Better describe this procedure.

Reviewer #2:

Remarks to the Author:

In this study, Zhou et al investigated the mechanisms of multisensory processing between auditory and olfactory cortex in humans, using intracranial EEG recordings. The participants were people with focal epilepsy who had electrodes implanted for planning of epilepsy surgery. The experimental paradigm was an auditory cue (e.g. the spoken word "mint"), followed by an odor 5 seconds later (e.g. the smell of mint), and then a participant response for whether the stimuli were congruent.

The main findings were (i) that the auditory cue produced a response in both auditory and olfactory cortex, with low-frequency phase synchrony between these regions, (ii) incorrect auditory-olfactory matching was associated with a low phase-locking value (PLV), and (iii) auditory low-frequency phase corresponds to modulation of olfactory beta amplitude. The authors' interpretation is that multisensory integration between auditory and olfactory stimuli requires phase synchrony between these sensory cortices.

Multisensory integration and the binding problem is certainly a topic of broad interest. The novel aspect of this study is the intracranial demonstration of auditory-olfactory phase synchrony in humans. As noted throughout the manuscript, the mechanism of phase synchrony for integration of sensory information is already a well-established hypothesis, with the most supporting evidence coming from animal studies and in non-olfactory modalities. In this context, the presented findings will be a valuable contribution to the olfactory and cognitive literature.

Main comments:

1) What were the selection criteria for participant inclusion in the study?

2) While this study clearly addresses multisensory processing and the interactions between auditory-olfactory sensory cortex, does it really examine multisensory integration? The findings are primarily based on the neural responses after the uni-modal auditory cue only, well before the odor is presented. This paradigm is quite different to a naturalistic cross-modal stimulus (this reviewer thinks of sizzling bacon) that would have nearly simultaneous auditory and olfactory components that can be unified into a coherent perceptual experience. With this in mind, an alternative framing of the paradigm and findings might be as predictive coding at the piriform cortex, a narrative which would give greater emphasis to top-down influences on olfactory cortex than has been given. A justification or comment on the relevance/difference between multisensory integration and predictive coding perspectives would be valuable.

3) That auditory-olfactory integration "requires" phase synchrony between these regions is a strong claim that needs further justification. The finding of low PLV for trials with incorrect responses partly supports the manuscript title, but the converse would be more convincing; i.e. can it be shown that trials with low PLV all lead to incorrect responses?

4) An important consideration is whether these patients with temporal lobe epilepsy are good representatives of normal auditory-olfactory processing. These patients are known to generally have a mild deficit on olfactory testing, with impaired activation of the piriform cortex seen on nuclear medicine and fMRI studies at the epileptic side, as well as atrophy of the piriform cortex in patients with hippocampal sclerosis. Epilepsy patients are perhaps the only setting where this study could be ethically conducted (for which I commend the authors) but the concurrent presence of disease pathology should be openly acknowledged as a potential limitation. Can the authors provide any data to reassure readers that the olfactory function of these participants was not too different to people without epilepsy? Did interictal epileptiform discharges occur during the experimental period, and were any steps taken to account for these in the EEG analysis?

5) All but one of the 7 patients had electrode implantation on the left hemisphere, which is relevant to the auditory cortex implantation at the (presumably) language-dominant hemisphere. Can the findings be justifiably generalized to auditory cortex bilaterally or are they only applicable in this paradigm to the language dominant side?

Other comments

-Abstract line 35: is the phrase "... the same temporal frame" intended in a technical sense? If so, defining this concept would be helpful to the general reader. The same applies to line 496 "phase resetting ... into the same temporal space".

- Abstract line 39 "trials when the odor and word were correctly matched" is ambiguous. Do the authors mean "trials where the auditory and olfactory stimuli were congruent", or "trials where the participant response was correct"?

- Subheading on line 372 "Low-frequency auditory cortical oscillations drive olfactory cortical gamma oscillations." . Should this read "... drive olfactory beta oscillations.." as is reported in the abstract ? The same applies to lines 518 and the following paragraph – should this discussion be of beta rather than gamma frequencies?

- Figure 6b & d : the third column, the contrast of "pre-post" means that an increase in comodulation corresponds to a negative value (blue). The inverse contrast "post-pre" may be more easily interpreted by the reader.

Reviewer #3:

Remarks to the Author:

In this manuscript by Zhou et al., subjects are requested to match the name of a smell (auditory cue) with the smell itself (olfactory cue) while intracranial EEG recordings are made from auditory and olfactory cortex. The authors report that: 1. The auditory cue elicits changes in neural activity in piriform cortex. 2. Correct matching of the auditory and olfactory cues is associated with phase synchronization between piriform and auditory cortical LFP. 3. Modulation of low frequencies in Auditory cortex is associated with modulation of the amplitude of high frequencies in piriform cortex. The authors suggest that these findings indicate a temporal timeframe for inter-regional (and inter-modal) integration of information and that this may serve for identification of odors by combining olfactory and other sensory inputs.

These are interesting findings that tackle the very important problem of multimodal integration and are in line with studies in other sensory systems and animal models. My main concerns relate to the relationship between the electrophysiological findings and the behavioral task and demands.

Major comments:

1. The task used in this study requires subjects to combine auditory and olfactory cues, however it is not clear that all of the effects are dependent on this integration. For instance, it is possible that passive listening and smelling to matching auditory and olfactory cues would account for some of the effects seen here. Would the same effect be seen if the same stimuli were presented and the task was purely odor-guided?
2. I think it is a little hard to relate the LFP findings to the behavioral requirements of the task with such limited analysis of the behavior itself. What is the success rate for each subject? Do they improve within the session? Are subjects better on match or non-match trials?

Minor comments:

1. In figures 2 and 6, it is not clear from the text and figure legends if various heat-maps show single subject data or are they averaged across subjects? If the latter, then it would be of interest to also see single subject heat-maps.
2. I think that a lot of the analysis could be made more accessible to a wider audience by adding some explanatory sentences explaining intuitively what is being extracted. What is "modulation index"? what is PLV?
3. The authors suggest in the discussion that their findings relate to "how the human brain enables odor identification by integrating multisensory cues" however in the task the auditory cue does not carry any information about the odors and is not involved in their identification.

Reviewers' comments:

Reviewer #1 (Remarks to the Author):

General Comments

The submission is well written and organized. The results are important and the methods and results are succinctly presented, but their description leaves the reader with much ambiguity. The results are clearly positive with respect to the hypotheses presented in the Intro. The auditory cue induces oscillations in the olfactory cortex prior to odor arrival. Evidence is also presented for long-range phase synchronization between auditory and olfactory cortices at this time; and there is evidence for phase-amplitude coupling between the low frequency (theta and delta) phase and the high frequency (beta and gamma) olfactory amplitude. This is true for low-frequency oscillations in both auditory and olfactory cortices.

We are very thankful for your helpful comments on our manuscript. Your thoughtful and detailed comments have resulted in a greatly improved manuscript.

Despite the positive results, particularly troubling is the lack of grid recordings for subjects 6 and 7 (Fig 1). Since their auditory recordings were different from the other subjects, how were their results compared to that of the others? Also, the question of hemispheric specialization is raised – each subject was implanted in the left hemisphere, where language is represented in most subjects, except for S7. The inclusion of S7, with right hemisphere implantation Table 1, may be confounding the results.

Response: Thank you for pointing this out. We agree with you that the difference in recordings for subjects 6 and 7 should be better addressed in the manuscript. These two subjects had clinical implantation schemes which were purely stereotactic—the surgeon implanted only depth wires; no grid was used. We included these subjects in our analyses because each had a depth wire electrode in the STG. Depth wires typically have 8 electrodes along each wire, and in these two cases, one of the more lateral electrodes was in STG, only several millimeters from the cortical surface, and close to the location of other subjects' grid electrodes. The locations of these depth wire electrodes are shown in Figure 1, for subjects 6 and 7.

In response to your comment, to ensure the signals we obtained from depth wire electrodes were comparable to those obtained from grid electrodes in STG, we compared the power spectral density of the depth recordings to the grid recordings. We found that the frequency composition of the two electrode types were very similar:

While the grid electrodes were on the surface of the brain, the particular electrodes that we analyzed on the depth wires were just below the surface, in the same region, namely STG. Because of this, we expected that they would show similar auditory responses. Indeed, we did find that auditory responses recorded with depth wires were indistinguishable from those recorded from grids both in terms of effect sizes and frequency composition (Supplementary Figure 1). See also figure below showing grid and depth spectrograms along with the average Z-score values within theta and gamma frequency ranges (panels to the right of the depth wire spectrograms):

The fact that we found similar results with depth wire electrodes and grid electrodes suggests high reproducibility of our results, and we believe this strengthens our findings.

The question of hemispheric specialization is a very good point, and was raised by another reviewer as well. Thank you for raising this point. We chose to analyze data from the right hemisphere in one participant based on several recent studies suggesting that while there is clear left dominance in language representation, the right hemisphere does also play a significant role (Lindell 2006). We agree with you that including data from the right hemisphere could potentially confuse the interpretation of the results. However, we feel that these potential problems are far out-weighted by the benefits of including an additional subject, given that iEEG subjects are rare. In considering inclusion of this subject's data, we noted that PLV findings from the right hemisphere electrode were consistent with findings from the left hemisphere electrodes, further

supporting our decision to include them in the paper. We have added a discussion of these points to the manuscript in the discussion on lines 456 - 464.

Lindell, Annukka K. 2006. "In Your Right Mind: Right Hemisphere Contributions to Language Processing and Production." *Neuropsychol Rev* 16 (3): 131–48. doi:10.1007/s11065-006-9011-9.

One problem is that the manuscript is too long. I suggest that, in order to shorten it, the confirmatory results for depth strength (PC vs PC-adjacent) and respiration (and related figures) be moved to a supplementary material section.

Response: Thank you for this helpful suggestion. We agree the manuscript is too long. Rather than placing these figures into a supplementary materials section, we have combined them into a single figure, and have shortened the text in which we discuss these results (see new Figure 3). We hope you think this helps. If not, we would be happy to place these findings into a supplement.

Specific Comments

Thank you for this detailed and thoughtful series of helpful comments and questions. We agreed with all of your comments and suggestions, and have made the requested changes in all cases. Specific answers to your questions are written out below.

79: Change "An increasing" to "A growing".

Response: Thank you, this change has been made.

80-82: It should be pointed out that this has been only a hypothesis before now - such patterns have been proposed to mediate ...

Response: Thank you- This is a helpful point. We have changed the sentence to "Coherent oscillatory firing patterns have been proposed to mediate ..."

90: "rodents and humans" - similar results have been seen in monkeys doing cognitive saccade tasks – add reference to Babapoor-Farrokhran et al. 2017 Nature Communications. Also add reference to Meehan et al. 2017 Brain Struct Funct where an auditory cue that predicted a visual stimulus led to effects in human primary visual cortex.

Response: Thank you for pointing out those important references that we missed, we have added them to our revised manuscript.

92-93: This hypothesis also assumes that the cue, preceding the sensory stimuli, affects processing of the ensuing stimulus in the sensory cortex. Cite the paper by Brovelli et al. 2004 PNAS as establishing that pre-stimulus oscillatory synchrony organizes information in primary sensory cortices.

Response: We agree that this is an important reference that established the role of

synchrony in multisensory integration in primary sensory cortices, which we have added to our revised manuscript.

95: *Phase synchrony is consistent with "communication" but does not establish that communication is enabled.*

Response: We have revised the sentence from "... prior work suggests that synchronized oscillations enable communication ..." to "... prior work suggests that synchronized oscillations reflect communication"

100: *"driving" is NOT measured here! See next comment.*

102: *Phase-amplitude coupling does NOT entail any sense of "driving". To have phase at one frequency coupled to amplitude at another frequency merely says that there is coupling, not that 1 drives the other.*

Response: Thank you for this important point. We agree that both phase locking value analysis and phase-amplitude coupling are measuring functional connectivity rather than causality between distinct brain regions. We have replaced the word "driving" with "facilitating/coupling".

135: *Why is there no grid for S6 & S7? Does "depth wire implanted" mean that depth wires were used in both auditory and olfactory systems for subjects S6 and S7? If so, where were the auditory electrodes implanted? Can their depth LFP recordings be compared with the ECoG recordings of the other subjects?*

Response: Thank you for this comment. We have partially addressed this question earlier in our response to your comments. The actual locations of these depth wire electrodes are shown in Figure 1. For these subjects, the clinical electrode implantation scheme didn't include any grid electrodes. However, the more lateral electrodes on the implanted depth wires were just below the surface of the brain, in the same region as the grid, namely STG. The electrodes on these depth wires used in our analysis were several millimeters below the surface. The results obtained from these depth wire electrodes were consistent with results from grid electrodes, as can be seen in the individual auditory spectrograms in the supplementary figure (Supplementary Figure 1) and also in individual phase locking value results (Figure 4 in revised manuscript).

146: *How do the spectrograms compare when the baseline is not removed? Recent LFP studies suggest that non-baseline-corrected spectrograms should also be examined.*

147: *How do the spectrograms compare when the baseline is not removed? Recent LFP studies suggest that non-baseline-corrected spectrograms should also be examined.*

Response: Thank you for pointing this out, we agree. We have computed the spectrogram from Figure 2a without baseline correction, and the effects are still clearly evident :

We have added a panel to Supplementary Figure 1 showing these spectrograms without baseline correction.

We would also point out to the reviewer that Figure 2d shows single-subject, single-trial, raw amplitude that has not been baseline corrected. This further indicates that baseline removal did not affect our findings.

152: Which PC electrode was used, and how was it selected?

Response: Thank you for pointing this out, we should have mentioned this in the manuscript. For each subject, we first determined which subset of electrodes was anatomically located inside piriform cortex. This typically included between 1 and 3 electrodes along the depth wire. For the subjects who had only a single electrode in piriform, we used that one. For subjects with multiple electrodes within PC, we chose the one that was closest to the center of piriform cortex. Notably, we did analyze all piriform electrodes separately with similar results. We have added this information, and a description of how we selected the STG electrode, to the methods section on lines 557-569.

156: The nature of this segmentation and its relevance are unclear. How long were the epochs? Is only 1 epoch displayed?

Response: We focused our analysis on the time window following auditory cues (from -0.55s prior to 1.5s following cue onset). Figure 2d shows the raw amplitude time series for all trials for each patient, computed using the Hilbert Transform. We have clarified this in the figure legend of Figure 2.

159: Is this the same as AUD shown in figure 2E? Was there even an electrode in STG for subjects 6 & 7?

Response: Thank you for catching this error, we have change "STG" to "AUD". Yes, there was an electrode in STG for these two subjects who had depth wires.

161-162: The description of panel F should be clarified. What is meant by the "real peak difference"? The peak difference should vary from trial to trial. Why is it represented by a single vertical line?

Response: We calculated the peak latency using the Z score time series from the spectrograms. More specifically, the spectrogram, similar to Figure 2a, was calculated for the specific frequency range (3-5 Hz). That means we extracted the amplitude time series of that specific frequency range and segmented it into epochs aligned to cue onset. Then we calculated the average over trials for STG and PC separately. The Z score of this average latency value was obtained by removing its baseline average, which was further divided by the standard deviation of the permuted baseline

distribution, resulting in one peak latency for the STG and one for the PC. The vertical line is the difference between those two peak latencies. The statistical significance of this “real” peak latency difference was tested using a permutation method. For each permutation, trial labels for STG and PC were switched randomly for each trial. The peak latency difference between permuted STG and PC was calculated in the same way described above. After repeating this procedure 10,000 times, we obtained a distribution (the histogram in Figure 2f) of permuted peak latency difference. Thus the “real peak difference” refers to the value resulting from real (non-permuted) data. We corrected the phrase “real peak difference” to “actual (non-permuted)” to clarify this.

171: Change “reflecting” to “presumably reflecting”.

Response: Change was made.

178: *It is not clear that task-related LFP alpha & beta suppression corroborates prior work on auditory resting oscillations.*

Response: Thank you for this comment. We have removed this sentence from the manuscript.

194: Change to “presumably reflecting”.

Response: Change was made.

232-233: How many depth wires were there for each patient? Did they all extend into the PC? This should be better explained - if each electrode went into the depth of the brain, how could electrode contacts be displayed on the cortical surface?

Response: Each participant typically had between 4 and 5 depth wires implanted into medial limbic structures, including piriform cortex, amygdala and hippocampus. Each depth wire has eight to ten electrodes along its length. While only one depth wire typically extends into PC, there are typically up to 3 electrodes on that depth wire within PC. For the two participants with depth wires passing through STG, the most lateral electrode on that wire was a few millimeters from the brain surface. In this figure, we projected those lateral electrodes onto the brain surface for display purposes. This has been clarified in the manuscript in the methods on lines 557 - 569.

295: There cannot be 2 panel Bs in Fig 5.

Response: Thank you pointing out our mistake. We have corrected the labels and corresponding text.

314-316: This sentence should be disentangled. The first part presents a result and the second a speculation. Also, functional connectivity is not equivalent to communication. I think the former is more appropriate here.

Response: We agree with you that functional connectivity is a more appropriate word. We changed the sentence to:

"Low-frequency oscillations have been suggested as a mechanism of functional connectivity across distributed neural networks. Based on this, we hypothesized that, following the cue and prior to the presentation of the odor, low frequency LFP

oscillations in auditory and olfactory cortices would become phase locked. "

333: Were "not matched" trials analyzed? Why or why not?

Response: Your comment, in addition to comments from reviewer 3, led us to realize that we used confusing language throughout the manuscript to refer to accuracy in the behavioral task in each trial; in particular, we failed to distinguish "correct trials" from "matched trials". We used the term "correctly matched" to denote when the subject correctly answered the question "does the odor match the audio cue", even when the two were not matched. In other words, we measured accuracy in the task by correct answers, regardless of whether the audio cue matched the odor. Correct answers included two conditions: when the odor matched the audio cue and the subject answered "yes", and when the odor did not match the audio cue and the subject answered "no". Correct and incorrect trials were defined based on accuracy only—regardless of whether the stimuli matched. We have modified the text to remove the term "matching", and now refer only to "correct" trials. We analyzed correct versus incorrect trials (regardless of matching or mismatching cues); we did not separately analyze trials in which the word and cue matched from trials in which the word and cue did not match.

We created additional confusion by failing to make clear the timing of our analysis in relation to the behavioral task. This is relevant to your question, because our PLV analysis was conducted following the auditory cue stimuli, but prior to the odor stimuli, thus prior to any knowledge of the correctness or incorrectness of the trial. At the time we analyzed PLV, conditions of correctness did not yet exist, as the odor had not yet been presented. By later correlating this analysis with correct versus incorrect trials, we found that on trials in which subjects produced an incorrect response following odor (regardless of whether the odor matched the word), PLV was not present between olfactory and auditory cortices prior to odor presentation. Thus, following the auditory cue, the presence of olfactory-auditory PLV predicted accurate performance on the task.

In every instance where we discuss this point, we have clarified the difference between correct and incorrect/matching and non-matching. (See lines 515-516; 942-943).

341: Should the text say "correctly matched"?

Response: Addressed in the above response.

361-362: It isn't clear how voice-selective auditory cortex was identified. Specify how the contacts were selected.

Response: Related to our response to your comment from line 152, we have added a detailed description of our STG selection procedure to the methods on lines 557-569.

364-369: The conclusions here, which are about the olfactory system DO NOT FOLLOW from the preceding text, which is about the auditory system.

Response: Thank you for noticing this typographical error. This sentence should have been the start of a new paragraph; we have made this correction.

372: All of the results presented in this paper have been undirected. Consequently, there is no concept of driving. Phase-amplitude coupling, also, is not directional and thus the word "drive" is inappropriate.

Response: We completely agree, thank you for this important point. We've modified the text to modify any suggestion that our effects were directed.

382: In A & C, right panels, the x-axis should be labeled. Also, the value of including the right panels is not clear. How were MI values derived?

Response: Thank you for pointing this out. We included the right panel with the intention to illustrate at which phase the higher frequency amplitude is maximized, since MI doesn't give us any information of the preferred phase of the modulating signal. We have added a sentence to the cross-frequency coupling analysis section of the methods to clarify our MI computation. See lines 682-684.

403: It is still debatable how strongly this theory is "established".

Response: Thank you for pointing this out, we agree.

We changed the sentence to:

"Coupling between the phase of low frequency oscillations and the amplitude of higher frequency oscillations is thought to be involved in cognitive processing, thus providing a means by which network coupling can impact local computations"

410: What is this? Phase-amplitude coupling?

Response: We forgot to define the abbreviation at its first appearance. This is phase-amplitude coupling, and we have revised to define the abbreviation "PAC".

438: Change "were" to "was".

Response: Change was made.

480: Remove "behavioral".

Response: "behavioral" was removed.

492: Change "For example" to "Specifically".

Response: Change was made.

510: Is there any indication from the present results what the separate roles might be of hierarchical convergence and phase synchrony of sensory areas?

Response: This is an interesting question. However, our results cannot speak to this question because we were not able to record signals from the whole brain, given clinical constraints.

515: Even though the finding may be in line with this idea, it is important to note that the present study did NOT study communication across networks.

Response: We have changed the sentence from "... as a means of communication across neural networks." to "... as a means of coupling across neural networks."

647: The third column is unlabelled. Also, is it the number of trials used in the analysis? If so, S3 only had 11 trials used. This is too small a number for individual-subject analyses.

Response: Thank you for pointing this out. This typographical error has been corrected. This column represents the final number of trials that were used for each participant. S3 completed 16 trials in total due to clinical constraints, and after removing bad trials, there were 11 remaining. However, even with only 11 trials, we observed strong single-trial amplitude increases in piriform cortex (see Figure 2d), and a strong response in both auditory cortex and piriform cortex, as shown by the spectrogram (figure below from Supplementary Figure 1). We didn't do individual phase locking value analysis for S3 because the estimation of phase locking value requires many more trials.

651: It's not clear how the surrogate events were generated. Was the analysis at the group level?

651: It is not clear what "circularly shifting" means.

Response: We computed spectrograms at both the group-level (Figures 2&3) and individual level (Supplementary Figure 1). To generate the surrogate events, a random number was added to all events (shifted in time), thus the relative distance between events was preserved. Circular shifting means that if the shifted event was out of range, the modulus of the length of the time series was used, which was done using Matlab's mod function (`mod(event_location, length_of_time_series)`). We have clarified this by revising corresponding text in methods section.

693: How were phase time series obtained? Wavelets? EMD? Hilbert Transform?

Response: Thank you for pointing this out. We calculated the analytic signal of the band-pass filtered signal using Hilbert transform. The phase time series, as well as amplitude time series, were obtained from this analytic signal. We clarified this by changing the sentence to:

" PLV is a measure of the consistency of the phase difference at a specific frequency and time between two regions over trials (Supplementary Figure 2). To obtain the phase time series, we band-pass filtered the raw time series between 1 and 30 Hz in 50 log-spaced frequencies (bandwidth: 2 Hz) for each region."

700-716: *Poor English grammar in several places.*

Response: We did a thorough grammar check for this paragraph and revised accordingly.

718-719: Give a better description of this method. What is MI?

Response: To better describe modulation index, we added the following sentence " MI measures the uniformity of the distribution of the amplitude of higher frequency

against the phase of low frequency." to the "Cross-frequency coupling analysis" section.

727-731: Give a more complete description of the procedure used.

Response: We have added a more complete description to this section of the methods.

768: This text is too terse! Better describe this procedure.

Response: We have added a better description of our method used to calculate the Z Score.

Reviewer #2 (Remarks to the Author):

In this study, Zhou et al investigated the mechanisms of multisensory processing between auditory and olfactory cortex in humans, using intracranial EEG recordings. The participants were people with focal epilepsy who had electrodes implanted for planning of epilepsy surgery. The experimental paradigm was an auditory cue (e.g. the spoken word "mint"), followed by an odor 5 seconds later (e.g. the smell of mint), and then a participant response for whether the stimuli were congruent.

The main findings were (i) that the auditory cue produced a response in both auditory and olfactory cortex, with low-frequency phase synchrony between these regions, (ii) incorrect auditory-olfactory matching was associated with a low phase-locking value (PLV), and (iii) auditory low-frequency phase corresponds to modulation of olfactory beta amplitude. The authors' interpretation is that multisensory integration between auditory and olfactory stimuli requires phase synchrony between these sensory cortices.

Multisensory integration and the binding problem is certainly a topic of broad interest. The novel aspect of this study is the intracranial demonstration of auditory-olfactory phase synchrony in humans. As noted throughout the manuscript, the mechanism of phase synchrony for integration of sensory information is already a well-established hypothesis, with the most supporting evidence coming from animal studies and in non-olfactory modalities. In this context, the presented findings will be a valuable contribution to the olfactory and cognitive literature.

Thank you for your thoughtful and helpful comments on our manuscript. You made some insightful comments which led us to run an additional analysis that has strengthened the findings.

Main comments:

1) *What were the selection criteria for participant inclusion in the study?*

Response: Thank you for pointing out this important information which was missing from the text. Inclusion criteria for this study included the following:

-clinical need for surgical implantation of electrodes into both olfactory and auditory cortices

- normal functioning sense of smell, as determined by self-report and the ability to accurately identify a series of common, familiar household odors, including the odorants used in this study.

-Any patient whose seizure focal zone included piriform cortex or a lesion in piriform cortex was excluded from the study

This information has been added to the methods section of the manuscript on lines 486-490.

2) While this study clearly addresses multisensory processing and the interactions between auditory-olfactory sensory cortex, does it really examine multisensory integration? The findings are primarily based on the neural responses after the uni-modal auditory cue only, well before the odor is presented. This paradigm is quite different to a naturalistic cross-modal stimulus (this reviewer thinks of sizzling bacon) that would have nearly simultaneous auditory and olfactory components that can be unified into a coherent perceptual experience. With this in mind, an alternative framing of the paradigm and findings might be as predictive coding at the piriform cortex, a narrative which would give greater emphasis to top-down influences on olfactory cortex than has been given. A justification or comment on the relevance/difference between multisensory integration and predictive coding perspectives would be valuable.

Response: We thank the reviewer for this insightful comment. We fully agree, and we had a difficult time deciding whether the data should be framed as multisensory integration or predictive coding. In fact, an earlier version of the manuscript was framed as cross-modal predictive coding. However, we modified this to discuss multisensory integration because of the strong interactions between the two sensory cortices, and the fact that our task requires integration of information from the two modalities for successful performance. However, we fully agree that a discussion of how our findings relate to predictive coding will be a valuable addition to the manuscript, and we have added a discussion of this to the manuscript on lines 384-398.

3) That auditory-olfactory integration "requires" phase synchrony between these regions is a strong claim that needs further justification. The finding of low PLV for trials with incorrect responses partly supports the manuscript title, but the converse would be more convincing; i.e. can it be shown that trials with low PLV all lead to incorrect responses?

Response: Thank you for this suggestion, this is a great idea! A challenge in doing what you suggested is that PLV is mathematically a cross-trial measure and cannot be estimated at the single-trial level, by definition. In other words, PLV can only be computed across a large number of trials that have been aligned to a particular event. Phase locking is then evident when the instantaneous phases of the single trials align similarly to each other, to the event. Because we need a large number of trials to compute the PLV, we tried the following analysis to address your suggestion: We pooled all trials (correct and incorrect) from all subjects into a single vector from which we randomly selected a subset of 71 trials, and we repeated this selection 10,000 times. The number 71 was chosen to exactly match the number of trials used in our main

correct versus incorrect analysis (see Figure 4c&d). On each repetition, we computed the PLV for the given random subset of trials, and we computed the proportion of trials within that subset that were correct. This allowed us to ask whether the PLV strength for a given subset of trials could predict the behavioral accuracy for that subset of trials. Results of this analysis were striking:

We found a strong correlation between PLV strength and performance for a given subset of trials. To ensure that this result was stable, we repeated this procedure 1,000 additional times, resulting in a mean correlation value of 0.9454 (95% confidence interval of [0.9407, 0.9502]).

This analysis will add strength to our findings, and we are thankful for your suggestion.

Results of this analysis have been added to the results section on lines 251-272, and to Figure 4.

4) An important consideration is whether these patients with temporal lobe epilepsy are good representatives of normal auditory-olfactory processing. These patients are known to generally have a mild deficit on olfactory testing, with impaired activation of the piriform cortex seen on nuclear medicine and fMRI studies at the epileptic side, as well as atrophy of the piriform cortex in patients with hippocampal sclerosis. Epilepsy patients are perhaps the only setting where this study could be ethically conducted (for which I commend the authors) but the concurrent presence of disease pathology should be openly acknowledged as a potential limitation. Can the authors provide any data to reassure readers that the olfactory function of these participants was not too different to people without epilepsy? Did interictal epileptiform discharges occur during the experimental period, and were any steps taken to account for these in the EEG analysis?

Response: This is a critical point, and we agree this should be better addressed in the manuscript. We have added more detail on the pathology of each of the participants. Most of them had normal MRIs with no medial temporal sclerosis. None of them had abnormal volume or functional activation of piriform cortex reported. All patients were able to reliably identify and distinguish between the odors.

Importantly, no participant had a seizure focus zone comprising auditory or olfactory cortex. Any trials during which epileptiform activity was present were removed from further analysis.

We have included a discussion of these points, and of the general limitations of studies conducted in this patient population, in the discussion on lines 442-464. See below:

One limitation of our study includes the use of patients with temporal lobe epilepsy. While human electrophysiology data can only be obtained from olfactory areas in

patients requiring this type of surgery, an important consideration is whether these patients are good representatives of normal auditory-olfactory processing. These patients sometimes show a mild deficit on olfactory testing, as well as potential atrophy of the piriform cortex in patients with medial temporal sclerosis. Surgical epilepsy patients are the only setting where this study could be ethically conducted, but we openly acknowledge that the disease pathology is a potential limitation. We took several steps to minimize the impact of these limitations on our data. No patient enrolled in our study had clinically noted abnormal volume or functional activation of piriform cortex. All patients were able to reliably identify and distinguish between the odors used in the task.

Importantly, no participant had a seizure focus zone comprising auditory or olfactory cortex. Finally, any trials during which epileptiform activity was present were removed from further analysis.

5) All but one of the 7 patients had electrode implantation on the left hemisphere, which is relevant to the auditory cortex implantation at the (presumably) language-dominant hemisphere. Can the findings be justifiably generalized to auditory cortex bilaterally or are they only applicable in this paradigm to the language dominant side?

Response: The question of hemispheric specialization is a very good point, and was raised by another reviewer as well. Thank you for raising this point. We chose to analyze data from the right hemisphere in one participant based on several recent studies suggesting that while there is clear left dominance in language representation, the right hemisphere does also play a significant role (Lindell 2006). We agree with you that including data from the right hemisphere could potentially confuse the interpretation of the results. However, we feel that these potential problems are far out-weighted by the benefits of including an additional subject, given that iEEG subjects are rare. In considering inclusion of this subject's data, we noted that PLV findings from the right hemisphere electrode were consistent with findings from the left hemisphere electrodes, further supporting our decision to include them in the paper. We have added a discussion of these points to the manuscript in the discussion on lines 456-464.

Regarding your question, "Can the findings be justifiably generalized to auditory cortex bilaterally or are they only applicable in this paradigm to the language dominant side?": Because we only had a single patient with an electrode on the right side, we cannot statistically compare results from the two hemispheres.

Other comments

-Abstract line 35: is the phrase "... the same temporal frame" intended in a technical sense? If so, defining this concept would be helpful to the general reader. The same applies to line 496 "phase resetting ... into the same temporal space".

Response: Thank you for this comment. We agree this phrase is not well-defined. We have removed it from the text in both locations in order to avoid confusion.

- Abstract line 39 "trials when the odor and word were correctly matched" is ambiguous. Do the authors mean "trials where the auditory and olfactory stimuli were congruent", or "trials where the participant response was correct"?

Response: Thank you for this comment. Your comment, in addition to comments from reviewers 1 and 3, led us to realize that we used confusing language throughout the manuscript to refer to accuracy in the behavioral task in each trial; in particular, we failed to distinguish “correct trials” from “matched trials”. We used the term “correctly matched” to denote when the subject correctly answered the question “does the odor match the audio cue”, even when the two were not matched. In other words, we measured accuracy in the task by correct answers, regardless of whether the audio cue matched the odor. Correct answers included two conditions: when the odor matched the audio cue and the subject answered “yes”, and when the odor did not match the audio cue and the subject answered “no”. Correct and incorrect trials were defined based on accuracy only—regardless of whether the stimuli matched. We have modified the text to remove the term “matching”, and now refer only to “correct” trials. We analyzed correct versus incorrect trials (regardless of matching or mismatching cues); we did not separately analyze trials in which the word and cue matched from trials in which the word and cue did not match.

We created additional confusion by failing to make clear the timing of our analysis in relation to the behavioral task. This is relevant to your question, because our PLV analysis was conducted following the auditory cue stimuli, but prior to the odor stimuli, thus prior to any knowledge of the correctness or incorrectness of the trial. At the time we analyzed PLV, conditions of correctness did not yet exist, as the odor had not yet been presented. By later correlating this analysis with correct versus incorrect trials, we found that on trials in which subjects produced an incorrect response following odor (regardless of whether the odor matched the word), PLV was not present between olfactory and auditory cortices prior to odor presentation. Thus, following the auditory cue, the presence of olfactory-auditory PLV predicted accurate performance on the task.

In every instance where we discuss this point, we have clarified the difference between correct and incorrect/matching and non-matching. (See lines 515-516; 942-943).

- Subheading on line 372 “Low-frequency auditory cortical oscillations drive olfactory cortical gamma oscillations.”. Should this read “... drive olfactory beta oscillations..” as is reported in the abstract? The same applies to lines 518 and the following paragraph – should this discussion be of beta rather than gamma frequencies?

Response: Thank you for pointing this out. We found coupling in both beta and gamma ranges. We have modified the abstract to state “high frequency oscillations”, we have modified the paragraph mentioned to clarify the language, and we have reduced the character count of the subheading to comply with Nature Communications formatting requirements, so that the word beta is no longer present.

- Figure 6b & d : the third column, the contrast of “pre-post” means that an increase in comodulation corresponds to a negative value (blue). The inverse contrast “post-pre” may be more easily interpreted by the reader.

Response: We agree with your point, and we made your recommended change (now Figure 5).

Reviewer #3 (Remarks to the Author):

In this manuscript by Zhou et al., subjects are requested to match the name of a smell (auditory cue) with the smell itself (olfactory cue) while intracranial EEG recordings are made from auditory and olfactory cortex. The authors report that: 1. The auditory cue elicits changes in neural activity in piriform cortex. 2. Correct matching of the auditory and olfactory cues is associated with phase synchronization between piriform and auditory cortical LFP. 3. Modulation of low frequencies in Auditory cortex is associated with modulation of the amplitude of high frequencies in piriform cortex.

The authors suggest that these findings indicate a temporal timeframe for inter-regional (and inter-modal) integration of information and that this may serve for identification of odors by combining olfactory and other sensory inputs.

These are interesting findings that tackle the very important problem of multimodal integration and are in line with studies in other sensory systems and animal models. My main concerns relate to the relationship between the electrophysiological findings and the behavioral task and demands.

Thank you for your insightful and helpful comments. Your review of our study has resulted in a stronger manuscript.

Major comments:

1. The task used in this study requires subjects to combine auditory and olfactory cues, however it is not clear that all of the effects are dependent on this integration. For instance, it is possible that passive listening and smelling to matching auditory and olfactory cues would account for some of the effects seen here. Would the same effect be seen if the same stimuli were presented and the task was purely odor-guided?

Response: Thank you for this insightful comment. While we cannot be certain that all of our effects are caused by integration of the auditory and olfactory cues, we do think it is very likely, since PLV was vastly reduced on trials when the information from the two modalities was not successfully linked. This suggests that when the auditory stimulus was not properly factored in to the olfactory decision, PLV was reduced (this is somewhat analogous to the situation you mention, with passive listening to auditory and olfactory cues, which does not require meaning about stimuli from each modality to be related to the other modality).

A related concern was brought up by reviewer 2, who mentioned the claim that integration requires phase synchrony would be made much stronger by showing that low PLV leads to incorrect responses. In response to reviewer 2's comment, we conducted an additional analysis which can speak to your concern as well. As you can see in this figure below (and see our response to Reviewer 2, comment 3), we found a striking relationship between PLV strength and performance on the task

We feel this additional analysis greatly strengthens the claims of the manuscript, and demonstrates a tight link between the LFP characteristics and successful integration.

We have given considerable thought to your question, “Would the same effect be seen if the same stimuli were presented and the task was purely odor-guided?”. If we understand this correctly (apologies if we’ve misinterpreted), you are suggesting that passive presentation of words and matching smells might produce PLV between piriform and auditory cortex, in the absence of any task.

Your comment, in addition to comments from reviewer 1, led us to realize that we used confusing language throughout the manuscript to refer to accuracy in the behavioral task in each trial; in particular, we failed to distinguish “correct trials” from “matched trials”. We used the term “correctly matched” to denote when the subject correctly answered the question “does the odor match the audio cue”, even when the two were not matched. In other words, we measured accuracy in the task by correct answers, regardless of whether the audio cue matched the odor. Correct answers included two conditions: when the odor matched the audio cue and the subject answered “yes”, and when the odor did not match the audio cue and the subject answered “no”. Correct and incorrect trials were defined based on accuracy only—regardless of whether the stimuli matched. We have modified the text to remove the term “matching”, and now refer only to “correct” trials. We analyzed correct versus incorrect trails (regardless of matching or mismatching cues); we did not separately analyze trials in which the word and cue matched from trials in which the word and cue did not match.

We created additional confusion by failing to make clear the timing of our analysis in relation to the behavioral task. This is relevant to your question, because our PLV analysis was conducted following the auditory cue stimuli, but prior to the odor stimuli, thus prior to any knowledge of the correctness or incorrectness of the trail. At the time we analyzed PLV, conditions of correctness did not yet exist, as the odor had not yet been presented. By later correlating this analysis with correct versus incorrect trails, we found that on trials in which subjects produced an incorrect response following odor (regardless of whether the odor matched the word), PLV was not present between olfactory and auditory cortices prior to odor presentation. Thus, following the auditory cue, the presence of olfactory-auditory PLV predicted accurate performance on the task.

In every instance where we discuss this point, we have clarified the difference between correct and incorrect/matching and non-matching. (See lines 515-516; 942-943).

2. I think it is a little hard to relate the LFP findings to the behavioral requirements of the task with such limited analysis of the behavior itself. What is the success rate for each subject? Do they improve within the session? Are subjects better on match or non-match trials?

Response: Thank you for pointing this out, we completely agree with you. We have added additional details of the behavioral results to the manuscript on lines 516-523. We now include the success rate of each individual participant. Participants did not improve over the course of the session, which was expected (Our experiment was not designed to induce learning). Subjects did perform better on trials when the word matched the odor than on trials when the word did not match the odor. However, as mentioned in response to your previous comment, we analyzed the LFP data prior to the time when the odor was presented, and hence prior to knowing whether the word and odor matched. Thus, we did not analyze trials of different matching status separately.

Further, as described in our response to your first comment, we have included an additional analysis that better relates the LFP findings to the behavioral data.

Minor comments:

1. In figures 2 and 6, it is not clear from the text and figure legends if various heat-maps show single subject data or are they averaged across subjects? If the latter, then it would be of interest to also see single subject heat-maps.

Response: Thank you for this comment. We computed the spectrograms and PLV analyses both at the group level and at the individual level. In Figures 2 and 6, (now Figure 5) the group level analyses are shown. We show individual-level heat-maps in Supplementary Figure 1. We also show individual participant, single-trial raw amplitude values in Figure 2d. For the modulation index analysis, however, we did not have enough trials of each condition (correct and incorrect) in each individual participant to perform individual level analyses. Therefore, the modulation index analysis was performed at the group-level only.

2. I think that a lot of the analysis could be made more accessible to a wider audience by adding some explanatory sentences explaining intuitively what is being extracted. What is “modulation index”? what is PLV?

Response: Thank you for pointing this out. We have added language to the manuscript in the methods section that provides a better description of the modulation index on lines 682-684.

We have also included a new figure (Supplementary Figure 2) to illustrate how PLV was calculated:

3. The authors suggest in the discussion that their findings relate to “how the human brain enables odor identification by integrating multisensory cues” however in the task the auditory cue does not carry any information about the odors and is not involved in their identification.

Response: We agree. And thank you for making this important point. While it is true that the auditory cue does not carry any predictive information about the odors, it does carry vital information necessary for proper performance of the olfactory perceptual decision. Thus, information from the auditory system is needed in order to make an olfactory perceptual decision. In this sense, integration is needed. We have included a discussion of your point in the discussion section of the manuscript on lines 373-378.

Reviewers' Comments:

Reviewer #1:

None

Reviewer #2:

Remarks to the Author:

Thank you to the authors, who have thoughtfully addressed each of the previous review comments. It is a well-written manuscript and an interesting and valuable contribution.

Reviewer #3:

Remarks to the Author:

I only have one (rather nit-picky) comment left and that relates to the behavioral analysis. All behavioral analysis should be performed on single subjects first and only then averaged. Similarly the relationship between PLV and performance should be performed on each subject individually and then averaged.

REVIEWERS' COMMENTS:

Reviewer #2 (Remarks to the Author):

Thank you to the authors, who have thoughtfully addressed each of the previous review comments. It is a well-written manuscript and an interesting and valuable contribution.

Thank you for this comment

Reviewer #3 (Remarks to the Author):

I only have one (rather nit-picky) comment left and that relates to the behavioral analysis.

All behavioral analysis should be performed on single subjects first and only then averaged. Similarly the relationship between PLV and performance should be performed on each subject individually and then averaged.

Thank you, we agree with you. We have reported all behavioral analyses at the single-subject level (see page 23, lines 527-530), and all PLV analyses at the single-subject level (see figures 2d, 3d, 4a-b). Regarding the relation between PLV and performance, there were not enough incorrect trials to perform this analysis separately in each participant, and therefore it was computed as a combined analysis.